# Build Roadmap for Automated Feature Transformation: A Graph-based Reinforcement Learning Approach

## Abstract

Feature transformation tasks aim to generate high-value features by combining existing ones through mathematical operations, which can improve the performance of downstream machine learning models. Current methods typically use iterative sequence generation, where exploration is guided by performance feedback from downstream tasks. However, these approaches fail to effectively utilize historical decision-making experiences and overlook potential relationships between generated features, thus limiting the flexibility of the exploration process. Additionally, the decision-making process lacks the ability to dynamically backtrack on efficient decisions, which hinders adaptability and reduces overall robustness and stability. To address these issues, we propose a novel framework that uses a graph to track the feature transformation process, where each node represents a transformation state. In this framework, three cascading agents sequentially select nodes and mathematical operations to generate new nodes. This strategy benefits from the graph structure's ability to store and reuse valuable transformations, and it incorporates backtracking via graph pruning techniques, allowing the framework to correct inefficient paths. To demonstrate the effectiveness and flexibility of our approach, we conducted extensive experiments and detailed case studies, demonstrating superior performance across a variety of datasets.

## 1 Introduction

Classic machine learning is highly dependent not only on the structure of the model but also on the quality of the training data (Sambasivan et al., 2021; Strickland, 2022; Borisov et al., 2022; Zha et al., 2023) (as depicted in Figure 1(a)). Traditionally, optimizing the dataset is referred to as feature engineering (Dong & Liu, 2018; Nargesian et al., 2017), which requires extensive manual intervention by domain experts (Conrad et al., 2022) and is time-consuming and labor-intensive. Other models, such as GBTs (Si et al., 2017) and deep neural networks (Bengio et al., 2013), can capture non-linear feature interactions spontaneously. However, they generally require significant amounts of data and computational power to achieve good generalization Grinsztajn et al. (2022), especially with limited tabular data Shwartz-Ziv & Armon (2022). Consequently, automated feature transformation has been proposed to adopt a data-centric perspective (Zha et al., 2023; Cui et al., 2024) to ensure both efficiency and automation.

**Background of Automated Feature Transformation:** The mainstream of existing automated feature transformation adopts an iterative perspective: *1) expansion-reduction approaches* (Kanter & Veeramachaneni, 2015; Khurana et al., 2016b; Horn et al., 2019) randomly combine and generate features through mathematical transformations, then employ feature selection techniques to isolate high-quality features. Those approaches are highly stochastic, lack stability, and are not optimization-oriented. *2) iterative-feedback approaches* (Tran et al., 2016; Li et al., 2023; Liu et al., 2024) aim to refine the feature space with the transformation towards reinforcement learning (Wang et al., 2022; Xiao et al., 2023a; 2024) and evolutionary algorithms (Khurana et al., 2018). Although those methods can optimize their strategies during the exploration, they discard the valuable experiences from historical sub-transformations and cannot backtrack on individual features. *3) AutoML approaches* (Zhu et al., 2022b; Zhang et al., 2023) partially adjust aforementioned issues by learning the pattern of the collected historical transformation records (Wang et al., 2024) thus reach a

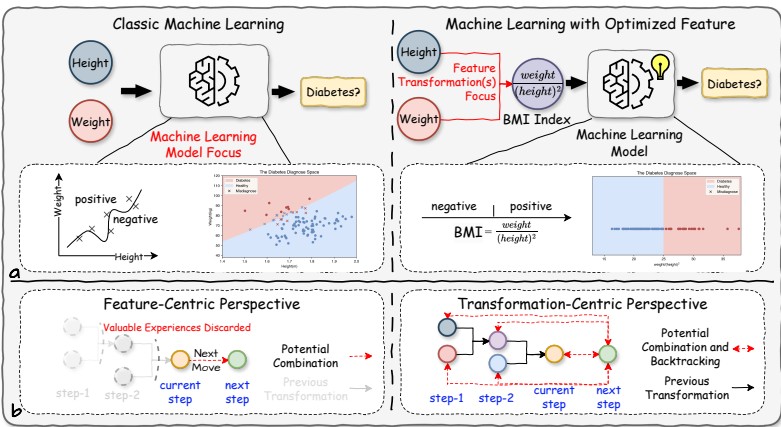

Figure 1: Motivation of this study. (a) Illustration of classic machine learning versus machine learning with optimized features in diabetes diagnosis.(b) A conceptual view of feature-centric and transformation-centric perspectives.

so-called global view of the action space. Nevertheless, a clear disadvantage of these methods is that they initially rely on the quality of collected transformations, which are essential for constructing a continuous search space that closely mirrors real-world conditions. After these discussions, a critical question emerges: *How to develop a framework that **maintains a global view**, **utilizes on the underlying connections between features**, and **dynamically adapts the transformation strategy**?*

**Our Perspective and Contribution:** In this work, we pivot to a transformation-centric approach in addressing the challenges outlined earlier (illustrated in the right section of Figure 1(b)). This shift brings forth three principal benefits that significantly enhance the capabilities of our reinforcement learning-based automated feature transformation framework: **(1) Enhanced Transformation Agility:** Our model is designed to capture and dynamically apply transformations across various stages of the feature transformation process rather than being restricted to transformations derived from the current feature set. This enables a more flexible and robust handling of features. **(2) Historical Insights Utilization:** We leverage deep learning techniques to extract and model latent correlations and mathematical characteristics from past transformation efforts. This historical insight informs our decision-making process, allowing the algorithm to execute transformation actions based on the lessons learned strategically. **(3) Robust Backtracking Mechanism:** Our approach incorporates a sophisticated backtracking system that utilizes historical transformation records for traceability. This feature ensures that the transformation process can revert or alter its course to avoid inefficient or suboptimal trajectories, thus optimizing the overall feature engineering pathway.

**Summary of Proposed Method: A Framework That Maintains Transformation Roadmap.** To capitalize on the benefits of a transformation-centric approach, we introduce the *Flexible Transformation-Centric Tabular Data Optimization Framework* (**TCTO**), an innovative automated feature transformation methodology employing a cascading multi-agent reinforcement learning (MARL) algorithm (Busoniu et al., 2008; Panait & Luke, 2005). Our framework is structured around an evolving feature-state transformation graph, which is maintained throughout the MARL process. This graph serves as a comprehensive roadmap, where each node and its path back to root node represents a unique sequence of transformations applied to the initial features of the dataset. Our optimization procedure comprises four steps: (1) clustering each node on the roadmap with mathematical and spectral characteristics, (2) state representation for each cluster, (3) cluster-level transformation decision generation based on multi-agent reinforcement learning; (4) evaluation and reward estimation for the generated outcomes. Iteratively, TCTO executes these steps while leveraging the traceability of the roadmap for a precise node-wise and step-wise pruning. This allows for targeted feature reduction and strategic rollbacks, optimizing the transformation pathway. Through rigorous experimental validation, we demonstrate that TCTO not only enhances the flexibility of the optimization process, but also delivers more resilient and effective results compared to traditional iterative optimization frameworks.

## 2 PRELIMINARY

### 2.1 IMPORTANT DEFINITIONS

**Dataset.** Formally, a dataset can be defined as $\mathcal{D} = [\mathcal{F}, Y]$, where $\mathcal{F} = \{f_1, \ldots, f_n\}$ represents $n$ features and $Y$ stands for the labels. Each row of $\mathcal{D}$ represents a single observation or data point, while each column corresponds to a specific attribute or feature of the observation.

**Operation Set.** To enhance the feature space and potentially improve the performance of downstream machine learning models, we can apply a set of mathematical operations to the existing features, generating new and informative-derived features. We define this collection of operations as the operation set, represented by the symbol $\mathcal{O}$. The operations[1] within this set can be categorized into two main types according to their computational properties: unary and binary operations *Unary operations* are those that operate on a single input feature, such as *square*, *exponentiation (exp)*, or *logarithm (log)*. *Binary operations* involve two input features and perform operations like *addition*, *multiplication*, or *subtraction*.

**Feature Transformation Roadmap.**
A feature transformation roadmap $\mathcal{G}$ is an evolving directed graph and could uniquely represent the global optimization process. Figure 2 shows an example of the new generation of edges and nodes. We can apply the roadmap to generate a new dataset $\mathcal{D}'$ with a given dataset, defined as $\mathcal{D}' = \mathcal{G}(\mathcal{D})$. This roadmap, denoted as $\mathcal{G} = \{V, E, \mathcal{A}\}$, consists of multiple tree structures where the number of trees equals the number of features

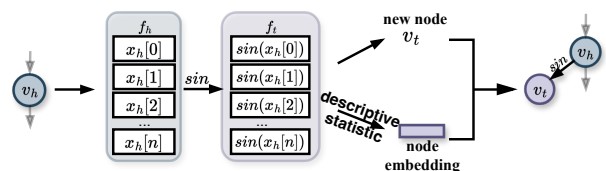

Figure 2: An example of feature transformation roadmap update: the feature $f_h$ conducts $sin$ operation generating the feature $f_t$. The embedding of node $v_t$ can be derived from the statistic description of generated feature $f_t$.

in the original dataset. $V = \{v_i\}_{i=1}^{m}$ and $E = \{e_i\}_{i=1}^{n}$ represent the set of feature state nodes[2] and transformation edges, respectively. $\mathcal{A}$ is the adjacency matrix. Each pair of nodes, connected by a directed edge, represents a new feature state $v_t$ generated from a previous state $v_h$ after undergoing the transformation represented by the type of edge $e$. The embedding of each node will be obtained via the descriptive statistics information (e.g., the standard deviation, minimum, maximum, and the first, second, and third quartile) of the generated features.

### 2.2 FEATURE TRANSFORMATION PROBLEM

As the toy model illustrated in Figure 1, given a downstream target ML model $\mathcal{M}$ (e.g., classification model, regression model, etc.) and a dataset $\mathcal{D} = [\mathcal{F}, Y]$, our objective is to find an optimal feature transformation roadmap $\mathcal{G}^*$ that can optimize the dataset through mathematical operation in $\mathcal{O}$. Formally, the objective function can be defined as:

$$\mathcal{G}^* = \arg\max_{\mathcal{G}} \mathcal{V}(\mathcal{M}(\mathcal{G}(\mathcal{F})), Y), \tag{1}$$

where $\mathcal{V}$ denotes the evaluation metrics according to the target downstream ML model $\mathcal{M}$.

## 3 PROPOSED METHOD

### 3.1 INSIGHTS OF THE PROPOSED METHOD

Figure 3 illustrates an overview of our proposed framework which comprises five key insights:

**Effective Transformation Action with Roadmap Clustering:** Previous study (Wang et al., 2022) shows that the mathematical operation between two distinct groups of features tends to generate high-informative features. In addition, a single feature transformation has little effect on downstream

---

[1]The detail of including mathematical operation can be found in Appendix A.2.5.

[2]Note that in the formulas, $v$ also represents the embedding of node $v$ for the sake of simplification.

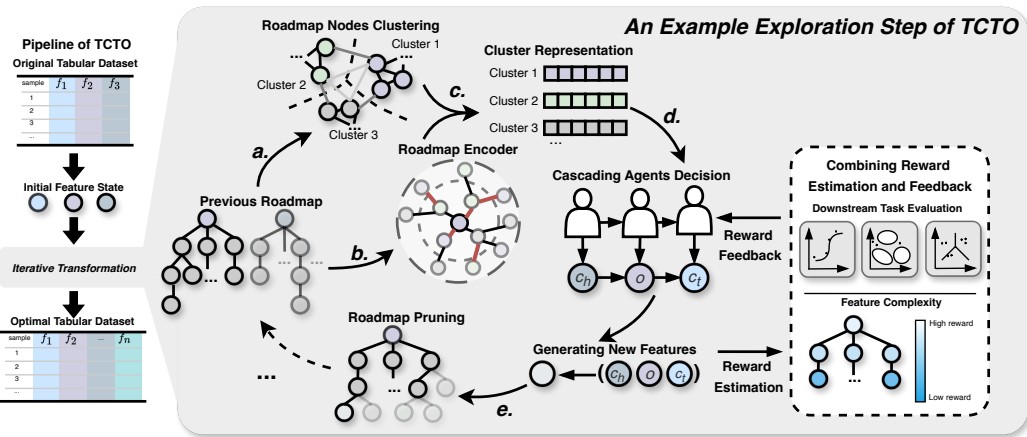

Figure 3: An overview of our framework: (a) cluster the nodes on roadmap; (b) represent the transformation roadmap; (c) calculate cluster representation; (d) reinforce multi-agent iterative feature transformation decision generation; (e) prune the roadmap effectively.

tasks' performance and hinders the optimization of reinforcement learning agents. Further, our insight into group-wise operation is that two close features will have similar historical transformation records or mathematical characteristics. With the roadmap accumulating, this latent relationship will reveal and could be critical to organizing effective yet efficient transformation.

**Roadmap-based State Representation for Each Agent:** Achieving an accurate state representation is crucial for enabling reinforcement agents to make informed decisions. In our framework, the transformation roadmap is a repository of intermediate transformation records, complete with their mathematical attributes. At each step, the agents select clusters of nodes, which can be seen as a subgraph on the roadmap. We then integrate a Relational Graph Convolutional Network (RGCN) (Schlichtkrull et al., 2018) to extract latent correlations within these historical records and capture the representation of each cluster. This approach allows our model to take advantage of the global insights gained from the RGCN, facilitating strategic transformation actions that are guided by the detailed state of the selected nodes.

**Multi-agent Reinforcement Learning based Transformation Decision**: Reinforcement learning has proven effective in addressing complex decision-making challenges across various domains. We employ three cascading agents that collaboratively construct unary and binary mathematical transformations. These agents operate sequentially to select the optimal head cluster, mathematical operation, and operand cluster, respectively. The chosen features undergo the specified mathematical operations, resulting in the generation of new features and the creation of new nodes within the roadmap. Additional details regarding the decision-making process will be provided in Section 3.3.

**Reward Estimation for Optimizing Agents:** Our model is optimized to generate high-quality features with minimal steps, enhancing efficiency. In this context, TCTO evaluates the generated features via the performance of downstream tasks to refine the reinforcement learning algorithm. In addition, we factored the complexity of the generated features into the reward function. This dual focus on performance and complexity ensures that the model aims for effectiveness while avoiding overly complex solutions that could hinder practical applicability and interpretability.

**Effective Roadmap Backtracking:** We have implemented two pruning strategies to manage the expanding complexity as the number of nodes in our roadmap grows. These approaches are designed to reduce the potential explosion in roadmap complexity, ensure the system remains efficient and manageable, and enhance our system's stability.

## 3.2 OPERATION ON DYNAMIC TRANSFORMATION ROADMAP

**Node Clustering on Roadmap**: As illustrated in Figure 4, we delineate the structure of roadmap $\mathcal{G}$ using its adjacency matrix $\mathcal{A}$, where each element $\mathcal{A}[i,j]$ quantifies the connectivity strength

between nodes $v_i$ and $v_j$. Each node in the roadmap is characterized by an embedding vector that encapsulates its feature information, denoted by the same notation $v$ for simplicity.

Inspired by Von Luxburg (2007), to enhance our analysis of inter-node relationships, we compute a similarity matrix $\tilde{\mathcal{A}}$ based on the cosine similarity between the embedding vectors of the nodes. The cosine similarity is calculated as follows:

$$\tilde{\mathcal{A}}[i,j] = \frac{v_i \cdot v_j}{\|v_i\|\|v_j\|} \quad (2)$$

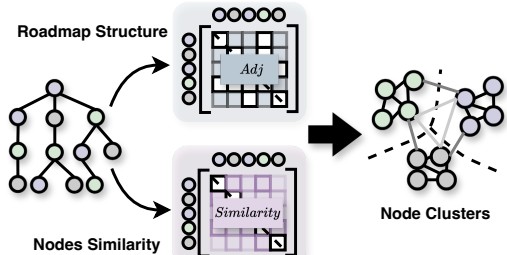

**Roadmap Structure**

*Adj*

**Nodes Similarity**

*Similarity*

**Node Clusters**

This similarity matrix $\tilde{\mathcal{A}}$ is integrated with the adjacency matrix $\mathcal{A}$ to amalgamate structural and feature-based information, thereby augmenting the efficacy of clustering or other graph analytical tasks. Furthermore, we define

Figure 4: Nodes clustering on roadmap based on structural and feature information.

an enhanced Laplacian matrix $\mathcal{S}$ to capture both structural and mathematical information from nodes, formulated as follows:

$$\mathcal{S} = \mathcal{D} - (\mathcal{A} + \tilde{\mathcal{A}}) \quad (3)$$

Here, $\mathcal{D}$ represents the degree matrix, with diagonal elements $\mathcal{D}[i,i]$ equal to the sum of the elements in the $i$-th row of $\mathcal{A} + \tilde{\mathcal{A}}$. The clustering module uses hierarchical clustering based on the eigenvalues and eigenvectors of $\mathcal{S}$ to identify the optimal roadmap partition into clusters. The clustering module treats each eigenvector corresponding to node $v_i$ as an initial singleton cluster and iteratively merges pairs of shortest clusters to progressively form larger clusters. This process continues until the cluster number reaches a specified setting, denoted as $k$. The set of clusters is denoted as $C = \{c_i\}_{i=1}^k$.

**Cluster State Representation with Roadmap**: As illustrated in Figure 5, we construct a dual-layer RGCN framework to disseminate and consolidate information across nodes, utilizing various relationship types to accurately represent the state of each cluster, described as:

$$v_i^{(l+1)} = \phi\left(\sum_{r=1}^{R} \sum_{j \in N_i^r} \frac{1}{c_{i,r}} W_r^{(l)} v_j^{(l)}\right) \quad (4)$$

where $v_i^{(l)}$ and $v_i^{(l+1)}$ represents the embedding of the $i$-th node in the roadmap at RGCN layer-$l$ and layer-$(l+1)$, respectively. $N_i^r$ denotes the set of neighboring nodes of $v_i$ with operation type $r$, and the degree normalization factor $c_{i,r}$ scales the influence of neighboring nodes. $r$ represents the relationships between nodes, which correspond to different

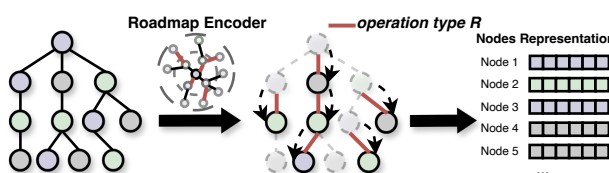

**Roadmap Encoder** — *operation type R*

**Nodes Representation**

Node 1
Node 2
Node 3
Node 4
Node 5
...

Figure 5: The roadmap encoding process utilizing graph convolution network.

operation types. The resulting sum is then passed through an activation function $\phi$ to produce the final representation of the node $v_i$. Based on the aggregated node representation, the representation of the cluster $c_i$ can be obtained by $Rep(c_i) = \frac{1}{|c_i|} \sum_{v \in c_i} v$, where $|c_i|$ denotes the number of nodes in cluster $c_i$.

**Roadmap Prune Strategy**: As illustrated in Figure 6, we employ two pruning strategies to ensure its stability during the feature transformation process.

*1) Node-wise pruning strategy:* entails the identification of $K$ nodes that show the greatest relevance to labels. This strategy computes the mutual information, defined as the relevance between each node's corresponding features and labels, as follows:

$$\mathcal{I}(v, Y) = \sum_{f_i \in v} \sum_{y_i \in Y} p(f_i, y_i) \log \frac{p(f_i, y_i)}{p(f_i)p(y_i)} \quad (5)$$

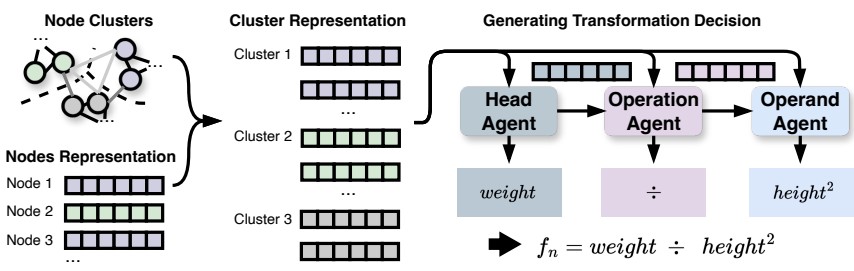

Figure 7: The reinforcement learning decision process. Three cascading agent cooperate to generate a binary transformation.

where $f_i$ denotes the element values of node $v$ and $y_i$ is its correlated label. $\mathcal{I}(v, Y)$ denoted the mutual information based score. $p(f)$ represents the marginal probability distribution, while $p(f, y)$ represents the joint probability distribution. Finally, the framework will select top-$K$ nodes by the score. The node-wise pruning strategy removes low-correlation nodes while preserving information as much as possible, ensuring exploration diversity.

*2) Step-wise backtracking strategy:* involves tracing back to the previous optimal transformation roadmap before the present episode to prevent deviating onto suboptimal paths. This stepwise backtracking ensures that the exploration process remains on the correct trajectory by revisiting and affirming the most effective roadmap configurations.

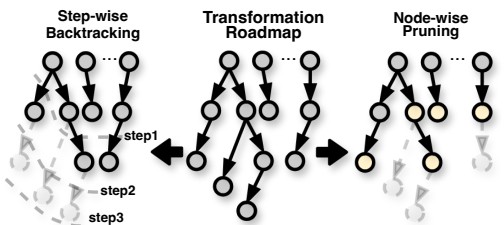

*3) When and how to prune the roadmap:* Pruning is recommended when the number of nodes reaches a set threshold. The node-wise pruning approach preserves diversity while minimizing complexity during the initial stages when

Figure 6: The two transformation roadmap pruning strategies.

agents are unfamiliar with the dataset. Once agents have grasped the fundamental policy, the step-wise backtracking strategy assumes leadership to enhance exploration stability. Combining both approaches, the agent explores a sufficiently large search space and maintains stable exploration in the later stages of training. Specifically, we adopt node-wise pruning in each step of the initial 30% of the exploration period, while the subsequent 70% is equipped with step-wise backtracking.

### 3.3 REINFORCEMENT LEARNING FRAMEWORK ON THE EVOLVING ROADMAP

**Cascading Reinforcement Learning Agents**: Figure 7 shows an example of the cascading agents' decision-making process. We utilize a series of cascading agents, each performing a specific task in sequential order. These agents collaborate in a step-by-step decision-making process, where the output of one agent serves as the input for the next. The first agent (head cluster agent) is responsible for selecting the head cluster, the second (operation agent) for choosing the most appropriate mathematical operation, and the third (operand cluster agent) for identifying the operand cluster. By using this cascading structure, each decision is informed by the context set by the previous agents, leading to a more efficient decision-making process. The details of each agent are as follows:

*1) Head Cluster Agent*: As described earlier, each node on the roadmap has been clustered into $\mathcal{C}$. The first agent aims to select the head cluster to be transformed according to the current state of each cluster. Specifically, the $i$-th cluster state is given as $Rep(c_i)$, and the overall state can be represented as $Rep(V)$. With the head policy network $\pi_h(\cdot)$, the score of select $c_i$ as the action can be estimated by: $s_i^h = \pi_h(Rep(c_i) \oplus Rep(V))$. We use $c_h$ to denote the selected cluster with the highest score.

*2) Operation Agent*: The operation agent aims to select the mathematical operation to be performed according to the overall roadmap and selected head cluster. The policy network in the operation

agent takes $Rep(c_h)$ and the global roadmap state as input, then chooses an optimal operation from the operation set $\mathcal{O}$: $o = \pi_o(Rep(c_h) \oplus Rep(V))$.

*3) Operand Cluster Agent*: If the operation agent selects a binary operation, the operand cluster agent will choose a tail cluster to perform the transformation. Similarly to the head agent, the policy network $\pi_t(\cdot)$ will take the state of the selected head cluster, the operation, the general roadmap state, and the $i$-th candidate tail cluster as input, given as $s_i^t = \pi_t(Rep(c_h) \oplus Rep(V) \oplus Rep(o) \oplus Rep(c_i))$, where $Rep(o)$ is a one-hot embedding for each operation. We use $c_t$ to denote the selected tail cluster with the highest score.

These aforementioned stages are referred to as one exploration step. Depending on the selected head cluster $c_h$, operation $o$, and optional operand cluster $c_t$, TCTO will cross each feature and then update the transformation roadmap (as shown in Figure 2 and Figure 7).

**Reward Estimation**: As illustrated in Figure 3, we reinforced and encouraged the cascading agents to conduct simple yet effective feature transformations. Based on this target, we employ the performance of downstream tasks and the complexity of the transformation roadmap as rewards to optimize the reinforcement learning framework, denoted as $\mathcal{R}_p$ and $\mathcal{R}_c$, respectively.

*(1) Performance of Downstream Tasks:* As the objective in Equation 1, $\mathcal{R}_p$ is calculated as follows:

$$\mathcal{R}_p = \mathcal{V}(\mathcal{M}(\mathcal{F}_{t+1}), Y) - \mathcal{V}(\mathcal{M}(\mathcal{F}_t), Y), \tag{6}$$

where $\mathcal{F}_t$ indicates the feature set at the $t$-th step.

*(2) Complexity of the Transformation:* The feature complexity reward $\mathcal{R}_c$ is defined as follows:

$$\mathcal{R}_c = \frac{1}{n} \sum_{j=1}^{n} \frac{1}{e^{h(v_j)}}, \tag{7}$$

where $h(v_j)$ represents the number of levels from the root node to node $v_j$ on $\mathcal{G}$. The total reward $\mathcal{R}$ is defined as follows: $\mathcal{R} = \mathcal{R}_p + \mathcal{R}_c$. In each step, the framework assigns the reward equally to each agent that has action.

**Optimization of the Pipeline**: In the cascading reinforcement learning setup described, the optimization policy is critical to refine the decision making capabilities of the agents involved: the Head Cluster Agent, Operation Agent, and Operand Cluster Agent. The overarching goal of this policy is to iteratively improve the actions taken by these agents to maximize the cumulative rewards derived from both the performance of downstream tasks and the complexity of transformations in the roadmap. The pseudo-code of cascading agents optimization and application phase are supplied in Algorithm 1 and Algorithm 2.

*1) Policy Optimization:* The learning process for each agent is driven by a reward mechanism that quantifies the effectiveness and efficiency of the transformations applied to the roadmap. Specifically, the optimization policy is framed within a value-based reinforcement learning approach, leveraging a dual network setup architecture: a prediction network and a target network. The prediction network generates action-value (Q-value) predictions that guide the agents' decision-making processes at each step. It evaluates the potential reward for each possible action given the current state, facilitating the selection of actions that are anticipated to yield the highest rewards. The target network serves as a stable benchmark for the prediction network and helps to calculate the expected future rewards. Decoupling the Q-value estimation from the target values is crucial to reducing overestimations and ensuring stable learning.

*2) Loss Function:* The loss function used for training the prediction network is defined as follows:

$$\mathcal{L} = \left( (\mathcal{Q}_p^\pi(s_t, a_t) - \left( \mathcal{R}_t + \gamma \cdot \max_{a_{t+1}} \mathcal{Q}_t^\pi(s_{t+1}, a_{t+1}) \right) \right)^2, \tag{8}$$

where prediction network $\mathcal{Q}_p^\pi(s_t, a_t)$ is the Q-value for the current state-action pair from the policy network $\pi(\cdot)$. $\mathcal{R}_t$ is the immediate reward received after taking action $a_t$ in state $s_t$, and $\gamma$ is the discount factor. $\max_{a_{t+1}} \mathcal{Q}_t^\pi(s_{t+1}, a_{t+1})$ is the maximum predicted Q value for the next state-action pair as estimated by the target network. The parameters of the prediction network are updated through gradient descent to minimize loss, thereby aligning the predicted Q values with the observed rewards plus the discounted future rewards. To maintain the stability of learning process, parameters of the target network are periodically updated by copying them from the prediction network.

Table 1: Overall performance comparison. 'C' for binary classification and 'R' for regression. The best results are highlighted in **bold**. The second-best results are highlighted in underline. (**Higher values indicate better performance.**) #Samp and #Feat denote the number of samples and features.

| Dataset | C/R | #Samp. | #Feat. | RDG | ERG | LDA | AFAT | NFS | TTG | GRFG | DIFER | FETCH | OpenFE | TCTO |
|---|---|---|---|---|---|---|---|---|---|---|---|---|---|---|
| Higgs Boson | C | 50000 | 28 | 0.695 | 0.702 | 0.513 | 0.697 | 0.691 | 0.699 | **0.709** | 0.669 | 0.697 | 0.702 | **0.709**$^{\pm 0.001}$ |
| Amazon Employee | C | 32769 | 9 | 0.932 | 0.934 | 0.916 | 0.930 | 0.932 | 0.933 | 0.935 | 0.929 | 0.928 | 0.931 | **0.936**$^{\pm 0.001}$ |
| PimaIndian | C | 768 | 8 | 0.760 | 0.761 | 0.638 | 0.765 | 0.749 | 0.745 | 0.823 | 0.760 | 0.774 | 0.744 | **0.850**$^{\pm 0.007}$ |
| SpectF | C | 267 | 44 | 0.760 | 0.757 | 0.665 | 0.792 | 0.760 | 0.760 | 0.907 | 0.766 | 0.760 | 0.760 | **0.950**$^{\pm 0.012}$ |
| SVMGuide3 | C | 1243 | 21 | 0.787 | 0.826 | 0.652 | 0.795 | 0.792 | 0.798 | 0.836 | 0.773 | 0.772 | 0.810 | **0.841**$^{\pm 0.012}$ |
| German Credit | C | 1001 | 24 | 0.680 | 0.740 | 0.639 | 0.683 | 0.687 | 0.645 | 0.745 | 0.656 | 0.591 | 0.706 | **0.768**$^{\pm 0.008}$ |
| Credit Default | C | 30000 | 25 | 0.805 | 0.803 | 0.743 | 0.804 | 0.801 | 0.798 | 0.807 | 0.796 | 0.747 | 0.802 | **0.808**$^{\pm 0.001}$ |
| Messidor_features | C | 1150 | 19 | 0.624 | 0.669 | 0.475 | 0.665 | 0.638 | 0.655 | 0.718 | 0.660 | 0.730 | 0.702 | **0.742**$^{\pm 0.003}$ |
| Wine Quality Red | C | 999 | 12 | 0.466 | 0.461 | 0.433 | 0.480 | 0.462 | 0.467 | 0.568 | 0.476 | 0.510 | 0.536 | **0.579**$^{\pm 0.003}$ |
| Wine Quality White | C | 4900 | 12 | 0.524 | 0.510 | 0.449 | 0.516 | 0.525 | 0.531 | 0.543 | 0.507 | 0.507 | 0.502 | **0.559**$^{\pm 0.003}$ |
| SpamBase | C | 4601 | 57 | 0.906 | 0.917 | 0.889 | 0.912 | 0.925 | 0.919 | 0.928 | 0.912 | 0.920 | 0.919 | **0.931**$^{\pm 0.002}$ |
| AP-omentum-ovary | C | 275 | 10936 | 0.832 | 0.814 | 0.658 | 0.830 | 0.832 | 0.758 | 0.868 | 0.833 | 0.865 | 0.813 | **0.888**$^{\pm 0.002}$ |
| Lymphography | C | 148 | 18 | 0.108 | 0.144 | 0.167 | 0.150 | 0.152 | 0.148 | 0.342 | 0.150 | 0.158 | 0.379 | **0.389**$^{\pm 0.016}$ |
| Ionosphere | C | 351 | 34 | 0.912 | 0.921 | 0.654 | 0.928 | 0.913 | 0.902 | **0.971** | 0.905 | 0.942 | 0.899 | **0.971**$^{\pm 0.001}$ |
| Housing Boston | R | 506 | 13 | 0.404 | 0.409 | 0.020 | 0.416 | 0.425 | 0.396 | 0.465 | 0.381 | 0.440 | 0.387 | **0.495**$^{\pm 0.015}$ |
| Airfoil | R | 1503 | 5 | 0.519 | 0.519 | 0.220 | 0.521 | 0.519 | 0.500 | 0.538 | 0.558 | 0.601 | 0.605 | **0.622**$^{\pm 0.011}$ |
| Openml_618 | R | 1000 | 50 | 0.472 | 0.561 | 0.052 | 0.472 | 0.473 | 0.467 | 0.589 | 0.408 | 0.565 | 0.393 | **0.600**$^{\pm 0.005}$ |
| Openml_589 | R | 1000 | 25 | 0.509 | 0.610 | 0.011 | 0.508 | 0.505 | 0.503 | 0.599 | 0.463 | 0.575 | 0.539 | **0.606**$^{\pm 0.003}$ |
| Openml_616 | R | 500 | 50 | 0.070 | 0.193 | 0.024 | 0.149 | 0.167 | 0.156 | 0.467 | 0.076 | 0.188 | 0.100 | **0.499**$^{\pm 0.052}$ |
| Openml_607 | R | 1000 | 50 | 0.521 | 0.555 | 0.107 | 0.516 | 0.519 | 0.522 | 0.640 | 0.476 | 0.571 | 0.430 | **0.670**$^{\pm 0.008}$ |
| Openml_620 | R | 1000 | 25 | 0.511 | 0.546 | 0.029 | 0.527 | 0.513 | 0.512 | 0.626 | 0.442 | 0.538 | 0.489 | **0.629**$^{\pm 0.001}$ |
| Openml_637 | R | 500 | 50 | 0.136 | 0.152 | 0.043 | 0.176 | 0.152 | 0.144 | 0.289 | 0.072 | 0.170 | 0.055 | **0.355**$^{\pm 0.022}$ |
| Openml_586 | R | 1000 | 25 | 0.568 | 0.624 | 0.110 | 0.543 | 0.544 | 0.544 | 0.650 | 0.482 | 0.611 | 0.512 | **0.689**$^{\pm 0.004}$ |

\* We report F1-score for classification tasks and 1-RAE for regression tasks.

\*\* The standard deviation is computed based on the results of 5 independent runs.

## 4 EXPERIMENT

We list the details of the experiment setting in the Appendix, where Appendix A.2.1 and A.2.3 introduce the platform information and the description of the dataset, all the methods compared and the preparation of the data are included in Appendix A.2.2. We also report hyperparameter settings and predefined mathematical operation set in Appendix A.2.4 and Appendix A.2.5. To thoroughly analyze the multiple characteristics of our approach, we also analyzed the running time complexity and bottleneck (Appendix A.3.1), space scalability (Appendix A.3.2), robustness (Appendix A.3.3), case studies (Appendix A.3.4), reward function (Appendix A.3.5) and scalability on large-scale datasets (Appendix A.3.6).

### 4.1 OVERALL COMPARISON

This experiment aims to answer the question: *Can our framework generate high-quality features to improve the downstream machine learning model?* Table 1 presents the overall comparison between our model and other models in terms of F1-score for classification tasks and 1-RAE for regression tasks. We observed that our model outperforms other baseline methods in most datasets. The primary reason is that it dynamically captures and applies transformations across various stages of the feature transformation process rather than being restricted to the latest nodes, thereby enhancing flexibility and robustness. Compared to expansion-reduction, our technique, along with other iterative-feedback methods, demonstrates a significant advantage in performance. The fundamental mechanism is that the reinforcement agent is capable of learning and refining its approach to the process, thereby achieving superior performance compared to random exploration. Another observation is that our model performs better than other iterative-feedback approaches, such as NFS, TTG, and GRFG. An explanation could be that our model identifies and incorporates hidden correlations and mathematical properties, enabling it to develop an improved strategy for feature transformation, drawing on extensive historical knowledge from previous efforts. Compared with the AutoML-based approach, DIFER, our technique demonstrates a significant improvement. This is primarily because DIFER relies on randomly generated transformations, which are unstable and prone to suboptimal results. Overall, this experiment demonstrates that TCTO is effective and robust across diverse datasets, underscoring its broad applicability for automated feature transformation tasks.

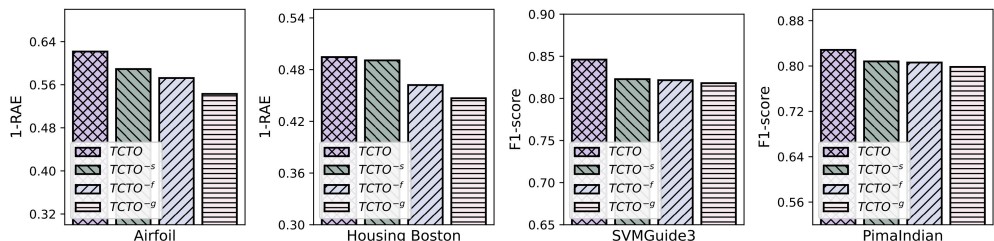

Figure 8: Comparison of TCTO and its variants in Regression and Classification tasks.

## 4.2 SIGNIFICANCE OF THE TRANSFORMATION ROADMAP

This experiment aims to answer the question: *How does the transformation roadmap impact each component in our model?* We design three different ablation variants: 1) $\mathbf{TCTO}^{-f}$ indicates that the clustering module ignores the mathematical characteristics. 2) $\mathbf{TCTO}^{-s}$ indicates that the clustering module ignores structural information. 3) $\mathbf{TCTO}^{-g}$ ablate the roadmap and adopt a feature-centric perspective. The comparison results of these variants are reported in Figure 8 and Figure 9.

**Impact on Clustering Component:** Figure 8 illustrates the effectiveness of the optimal features produced by our model and its variants in downstream tasks on the test dataset. Firstly, we discovered that TCTO against the other three variants, while $\text{TCTO}^{-g}$ showed the weakest performance. This indicates that the integration of roadmap structure and feature information is vital for a precise clustering, which can help the agents to organize transformation between two distinct groups of features, thus generating high-value features. We can also observe that $\text{TCTO}^{-s}$ outperforms $\text{TCTO}^{-f}$ on each task and dataset. This observation shows that $\text{TCTO}^{-f}$ is superior to $\text{TCTO}^{-g}$, i.e., the mathematical characteristic of the generated feature seems to be more significant than structural information. The underlying driver is that structural information from historical transformation can enhance the clustering component, thus resulting in better performance.

**Impact on Cluster State Representation:** From Figure 8, we can observe a decrease in the performance of downstream tasks when the roadmap structure is excluded, i.e., $\text{TCTO}^{-g}$. This performance decline is attributed to the loss of essential information that the transformation roadmap maintained. In contrast, utilizing the roadmap can enable agents to make strategic decisions based on comprehensive historical insights and complex feature interactions.

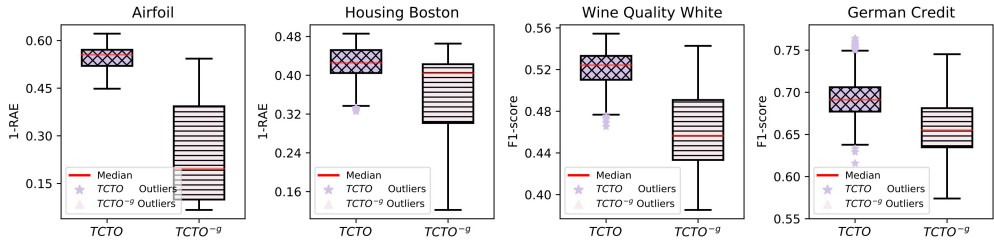

Figure 9: Stability comparison of TCTO and $\text{TCTO}^{-g}$ in four different datasets.

**Impact on Exploration Stability:** To assess stability, we collected the performance of the downstream task at each exploration step of TCTO and the ablation variation method $\text{TCTO}^{-g}$. Figure 9 displays box plots summarizing the distributional characteristics of the experimental results. We can first observe that the median line of our model is consistently higher than $\text{TCTO}^{-g}$. Additionally, the interquartile range (IQR), depicted by the length of the box, indicates that our model's performance distribution is more concentrated than the ablation variation. The observed stability in our model can be attributed to two primary factors. Firstly, the incorporation of historical and feature information within the roadmap provides guidance, steering the model towards more stable exploration directions. Secondly, the implementation of a roadmap pruning strategy alongside a backtracking mechanism plays a crucial role; it eliminates ineffective transformed features or reverts the model to the optimal state of the current episode, thereby ensuring stability throughout the exploration process.

### 4.3 ANALYSIS ON GRAPH PRUNING TECHNIQUE

This experiment aims to answer the question: *What is the impact of node-wise and step-wise pruning ratios?* To validate the pruning ratio sensitivity of our model, we set the ratio from 0 to 1 to observe the differences. We report the performance variations on Airfoil (regression task) and PimaIndia (classification task) in Figure 10. We observe that adopting more node-wise pruning, downstream ML performance improves initially and then declines. A possible reason is that the node-wise pruning could preserve search space diversity when agents are unfamiliar with the dataset. However, with more application of the node-wise pruning strategy, TCTO cannot backtrack to the previous optimal transformation roadmap, resulting in suboptimal paths and decreased performance. We set the node-wise pruning ratio to 30% according to the experimental results.

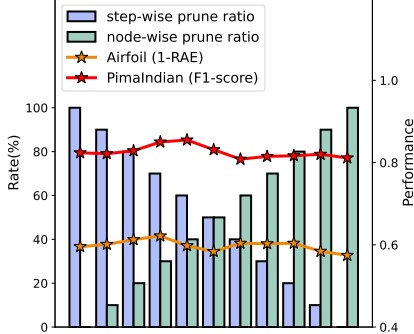

Figure 10: Study of the node-wise and step-wise pruning ratio on Airfoil and PimaIndia datasets.

## 5 RELATED WORK

Feature engineering refers to the process of handling and transforming raw features to better suit the needs of machine learning algorithms (Hancock & Khoshgoftaar, 2020; Chen et al., 2021). Automated feature engineering implies that machines autonomously perform this task without the need for human prior knowledge(Lam et al., 2017). There are three mainstream approaches: The expansion-reduction based method (Kanter & Veeramachaneni, 2015; Horn et al., 2019; Khurana et al., 2016b; Lam et al., 2017; Khurana et al., 2016a), characterized by its greedy or random expansion of the feature space(Katz et al., 2016; Dor & Reich, 2012), presents challenges in generating intricate features, consequently leading to a restricted feature space. The iterative-feedback approach (Khurana et al., 2018; Tran et al., 2016; Wang et al., 2022; Xiao et al., 2023a; Zhu et al., 2022a; Xiao et al., 2024) methods integrate feature generation and selection stages into one stage learning process, and aims to learn transformation strategy through evolutionary or reinforcement learning algorithms (Ren et al., 2023). However, these methods usually model the feature generation task as a sequence generation problem, ignoring historical and interactive information during the transformation progress, result in lack of stability and flexibility. The AutoML-based approaches (Wang et al., 2021; Zhu et al., 2022b; Xiao et al., 2023b; Ying et al., 2023) have recently achieved significant advancement. However, they are limited by the quality of the collected transformation and also the lack of stability and traceability during the generation phase. To overcome these problems, TCTO introduces a novel framework that integrates structural insights based on roadmaps and a backtracking mechanism with deep reinforcement learning techniques to improve feature engineering.

## 6 CONCLUSION REMARKS

We introduce TCTO, an automated framework for feature transformation. Our approach focuses on managing feature modifications through a transformation roadmap, which keeps track of and organizes the process to ensure optimal feature generation. There are three main benefits to our approach: (1) Preserving Transformation Records: The roadmap structure automatically logs all feature transformations, making it accurate to cluster similar features and enhancing the model's capabilities. (2) Insightful Decision-Making: By utilizing unique structural and mathematical characteristics, our cascading agents can make better decisions based on detailed state representations. (3) Increased Robustness through Backtracking: The roadmap's built-in backtracking feature allows the framework to correct or change its path if it encounters inefficient or suboptimal transformations, thereby improving the model's robustness and adaptability. Extensive experiments show that TCTO is effective and flexible in optimizing data for a wide range of applications. Further discussion including future work and application scenario is listed in Appendix A.4.

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

# A APPENDIX

## A.1 PSEUDO-CODE FOR PROPOSED METHOD

This section intends to provide the details of our methodology. Algorithm 1 describes the optimization process of our method. Through continuous interaction with the environment, the RGCN can effectively encode the transformation roadmap. Cascading agents are capability to make optimal decisions based on the current state.

---

**Algorithm 1** Cascading Agents Optimization Phase

---

**Input**: dataset $\mathcal{D}[\mathcal{F}, Y]$

**Initialization**: downstream ML task $\mathcal{M}$, evaluation metric $\mathcal{V}$, complexity metric $\mathcal{U}$, head cluster agent $\pi_h$, operation agent $\pi_o$, operand cluster agent $\pi_t$, cluster state representation $Rep(\cdot)$, reward $\mathcal{R}$

**Parameter**: training episode T, training step N, prune strategy node number p

**Output**: head cluster agent $\pi_h$, operation agent $\pi_o$, operand cluster agent $\pi_t$, RGCN encoder $Rep(\cdot)$

```
 1: for i = 0 to T do
 2:     G ← D                                          # 1. Initialize roadmap based on dataset
 3:     for j = 0 to N do
 4:         C ← clustering roadmap G
 5:         c_h ← π_h(Rep(G), Rep(C))                          # 2. Head agent decision
 6:         o ← π_o(Rep(G), Rep(C), Rep(c_h))               # 3. Operation agent decision
 7:         if o is a unary operation then
 8:             D ← (D ∪ (o → c_h))           # 4. Generate new features and update dataset
 9:         else
10:             c_t ← π_t(Rep(G), Rep(C), Rep(c_h), Rep(o))        # 5. Operand agent decision
11:             D ← (D ∪ (c_h → o → c_t))       # 6. Generate new features and update dataset
12:         end if
13:         G   UPDATES
14:         R ← R_p + R_c                                  # 7. Calculate combining reward
15:         Optimize π_h, π_o, π_t and encoder Rep based on R
16:         if node number of G > p then
17:             G ← Prune G
18:         end if
19:     end for
20: end for
21: return π_h, π_o, π_t, Rep
```

---

Algorithm 2 describes the application process of TCTO. The optimized reinforcement learning agents can make decisions based on the current roadmap, resulting in a dataset with better performance on downstream tasks.

## A.2 EXPERIMENT SETTINGS

### A.2.1 EXPERIMENTAL PLATFORM INFORMATION

All experiments were conducted on the Ubuntu 18.04.6 LTS operating system, AMD EPYC 7742 CPU, and 8 NVIDIA A100 GPUs, with the framework of Python 3.8.18 and PyTorch 2.2.0 Paszke et al. (2019).

### A.2.2 BASELINE METHODS AND DATA PREPARATION

We conducted a comparative evaluation of TCTO against seven other feature generation methods: (1) **RDG** randomly selects an operation and applies it to various features to generate new transformed features. (2) **ERG** conducts operations on all features simultaneously and selects the most discriminative ones as the generated features. (3) **LDA** (Blei et al., 2003) is a classic method based on matrix decomposition that preserves crucial features while discarding irrelevant ones. (4)

---

**Algorithm 2** Cascading Agents Application Phase

---

**Input**: dataset $\mathcal{D}[\mathcal{F}, Y]$, head cluster agent $\pi_h$, operation agent $\pi_o$, operand cluster agent $\pi_t$, cluster state representation $Rep(\cdot)$
**Initialization**: downstream ML task $\mathcal{M}$, evaluation metric $\mathcal{V}$, complexity metric $\mathcal{U}$
**Parameter**: testing episode T, testing step N, prune strategy node number p
**Output**: optimal dataset $\mathcal{D}'$

1: **for** $i = 0$ to $T$ **do**
2:     $\mathcal{G} \leftarrow \mathcal{D}$                                              # 1. Initialize roadmap based on dataset
3:     Let best performance $m = 0$, current best dataset $D' = D$.
4:     **for** $j = 0$ to $N$ **do**
5:         $\mathcal{C} \leftarrow$ clustering roadmap $\mathcal{G}$
6:         $c_h \leftarrow \pi_h(Rep(\mathcal{G}), Rep(C))$                        # 2. Head agent decision
7:         $o \leftarrow \pi_o(Rep(\mathcal{G}), Rep(C), Rep(c_h))$              # 3. Operation agent decision
8:         **if** $o$ is a unary operation **then**
9:             $\mathcal{D} \leftarrow (\mathcal{D} \cup (o \rightarrow c_h))$              # 4. Generate new features
10:         **else**
11:             $c_t \leftarrow \pi_o(Rep(\mathcal{G}), Rep(C), Rep(c_h), Rep(o))$      # 5. Tail agent decision
12:             $\mathcal{D} \leftarrow (\mathcal{D} \cup (c_h \rightarrow o \rightarrow c_t))$          # 6. Generate new features
13:         **end if**
14:         $\mathcal{G}$    UPDATES
15:         **if** $\mathcal{V}(\mathcal{M}(\mathcal{D})) > m$ **then**
16:             $m \leftarrow \mathcal{V}(\mathcal{M}(\mathcal{D}))$
17:             $D' \leftarrow D$                                          # 7. Update optimal dataset
18:         **end if**
19:         **if** node number of $\mathcal{G} > $ p **then**
20:             $\mathcal{G} \leftarrow Prune \, \mathcal{G}$
21:         **end if**
22:     **end for**
23: **end for**
24: **return** $\mathcal{D}'$

---

**AFAT** (Horn et al., 2019) overcomes the limitations of ERG by generating features multiple times and selecting them in multiple steps. (5) **NFS** (Chen et al., 2019) conceptualizes feature transformation as sequence generation and optimizes it using reinforcement learning. (6) **TTG** (Khurana et al., 2018) formulates the transformation process as a graph construction problem at the dataset level to identify optimal transformations. (7) **GRFG** (Xiao et al., 2024) employs a cascading reinforcement learning structure to select features and operations, which ultimately generates new discriminative characteristics. (8) **FETCH** (Li et al., 2023) is an RL-based end-to-end method that employs a single agent to observe the tabular state and make decisions sequentially based on its policy. (9) **OpenFE** (Zhang et al., 2023) is an efficient method that initially evaluates the incremental performance of generated features and then prunes candidate features in a coarse-to-fine manner.

To ensure experimental integrity, the datasets were divided into training and testing subsets to prevent data leakage. The training dataset, comprising 80% of the data, was used to optimize the reinforcement learning process. The testing datasets were used to evaluate the transformation and generation capabilities of the models. The partitioning principle was stratified sampling, which follows the same settings as in previous research Wang et al. (2022); Zhu et al. (2022b). Specifically, for regression tasks, we divided the labels into five ranges based on value size and randomly selected 20% from each range for testing, with the remaining portion used for training. For classification tasks, we selected 20% from each class for testing, with the remaining data used for exploration. In the model's final evaluation phase, we used the sci-kit-learn toolkit to test the on-hold generated dataset in downstream tasks and applied the n-fold cross-validation method provided by the toolkit to partition the data for testing. Downstream machine learning tasks were performed using Random Forest Regressor and Random Forest Classifier.

### A.2.3 DATASET AND EVALUATION METRICS

Table 1 provides a succinct summary of these datasets, detailing sample sizes, feature dimensions, and task categories. The datasets utilized for training our model were obtained from publicly accessible repositories, including Kaggle, LibSVM, OpenML, and the UCI Machine Learning Repository. Specifically, the details of the dataset source are listed below:

- LibSVM (Chih-Jen, 2022): SVMGuide3
- Kaggle (Howard, 2022): Amazon Employee
- UCIrvine (Public, 2022b): Higgs Boson, PimaIndian, SpectF, German Credit, Credit Default, Messidor_features, Wine Quality Red, Wine Quality White, SpamBase, Lymphography, Ionosphere, Housing Boston, Airfoil
- OpenML (Public, 2022a): AP-omentum-ovary, Openml_618, Openml_589, Openml_616, Openml_607, Openml_620, Openml_637, Openml_586

Our experimental analysis incorporated 14 classification datasets and 9 regression datasets. For evaluation, we utilized the F1-score for classification tasks and the 1-Relative Absolute Error (1-RAE) for regression tasks. In both cases, a higher value of the evaluation metric indicates that the generated features are more discriminative and effective.

### A.2.4 HYPERPARAMETER SETTINGS AND REPRODUCIBILITY

To comprehensively explore the feature space, we conducted exploration training for 50 episodes, each consisting of 100 steps, during the reinforcement learning agent optimization phase. Following optimizing, we assessed the exploration ability of the cascading agents by conducting 10 application episodes, each comprising 100 steps. Following existing research Wang et al. (2024); Xiao et al. (2024), we set the number of clusters $k$ to the square root of the current number of nodes, while the number of nodes triggered for pruning $K$ is set to four times the original number of features. During step-wise pruning, we utilize the $k$ most importance features. We utilized a two-layer RGCN as the encoder for the transformation roadmap, and an embedding layer for the operation encoder. The hidden state sizes for the roadmap encoder and operation encoder were set to 32 and 64, respectively. Each agent was equipped with a two-layer feed-forward network for the predictor, with a hidden size of 100. The target network was updated every 10 exploration steps by copying parameters from the prediction network. To train the cascading agents, we set the memory buffer to 16 and the batch size to 8, with a learning rate of 0.01. For the first 30% epochs, we employed a node-wise pruning strategy to eliminate low-quality features. Subsequently, we utilized a step-wise backtracking strategy for the remaining epochs to restore the optimal roadmap.

### A.2.5 MATHEMATICAL OPERATION SET

The operation set includes elementary unary and binary mathematical operation. For enhancing the transformation agility, we utilize some functional mathematical operation. The details of operation set are listed as follows. The token $x$ is a scalar, which implies each element in vector $X$.

- Elementary mathematical operation
  - Unary: $x^2, x^3, \sqrt{x}, \sin x, \cos x, \log_e(x), e^x$
  - Binary: $+, -, \times, \div$
- Functional mathematical operation
  - tanh: $x' = \frac{e^x - e^{-x}}{e^x + e^{-x}}$
  - sigmoid: $x' = \frac{1}{1+e^{-x}}$
  - reciprocal: $x' = \frac{1}{x}$
  - stand_scaler: $X' = \frac{X-\mu}{\sigma}$
    Note: $\mu$ and $\sigma$ is the mean and standard deviation of $X$, respectively.
  - minmax_scaler: $X' = \frac{X-X_{min}}{X_{max}-X_{min}}$
    Note: $X_{min}$ and $X_{max}$ mean the minus and max element of $X$, respectively.
  - quantile_transform: $X' = Quantile(X)$
    Note: Quantile transforms features to follow a uniform distribution.

### A.3 SUPPLEMENTARY EXPERIMENT

For analyzing the multiple characteristics of TCTO, we conducted supplementary experiments. We provide the runtime bottleneck analysis (results shown in Figure 11), the space complexity analysis (results shown in Table 3), the study of robustness (results shown in Table 4), case study on generated features (results shown in Table 5), weight of reward function (results shown in Figure 12) and scalability on large-scale datasets (results shown in Table 6).

#### A.3.1 RUNTIME COMPLEXITY AND BOTTLENECK ANALYSIS

This experiment aims to answer: *What is the main temporal bottleneck of* TCTO*?*

**Time Complexity Analysis:** Table 2 shows the analysis of the time complexity. Regarding of *clustering process*: It involves calculating the eigenvalue and eigenvector of a matrix, whose dimension depends on the feature number. The normal time complexity of eigen decomposition is $\mathcal{O}(m^3)$. Regarding of *decision process*: It is a neural network forward inference process. The time complexity lies on the latent dimension, neural network architecture and etc. We don't analysis it deeply. Regard-

Table 2: The time complexity analysis

| Time Complexity Analysis | | | | |
|---|---|---|---|---|
| Cluster | Decision | Update | Prune | Downstream task |
| $\mathcal{O}(m^3)$ | $R$ | $\mathcal{O}(mk)$ | $\mathcal{O}(nm)$ | $\mathcal{O}(Tmn\log n + T\log n)$ |
| $n$ :sample number | | $T$:the number of constructed trees during Random Forest algorithm | | |
| $m$:feature number | | $R$:related to the latent dimension, neural network architecture | | |
| $k$: generative feature number | | | | |

ing of *roadmap updating process*: In order to avoid the repeat nodes adding to the roadmap, we compare generative nodes with existing ones. Suppose the generative feature number is $k$, the time complexity of updating is $\mathcal{O}(mk)$. Regarding of *roadmap pruning process*: For node-wise pruning strategy, we select the most effective nodes from existing nodes based on their importance. The time complexity of calculating importance is $\mathcal{O}(nm)$. For step-wise pruning strategy, the time complexity is $\mathcal{O}(1)$ Regarding of *downstream task process*: Take random forest algorithm as an example, the constructing and training trees involves $\mathcal{O}(Tmn\log n)$ and testing model involves $\mathcal{O}(T\log n)$ time complexity. It is worth noting that although clustering process has $\mathcal{O}(m^3)$ time complexity, the calculating process of eigen decomposition is fast during empirical running time.

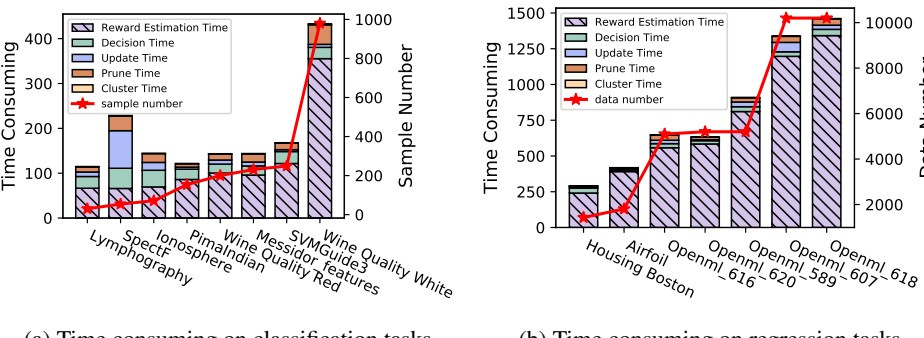

(a) Time consuming on classification tasks      (b) Time consuming on regression tasks

Figure 11: Time consumption of TCTO on different tasks.

**Runtime Bottleneck Analysis:** Figure 11 visualized the average empirical running time consumption on each dataset of different modules to analyze the time complexity, including reward estimation, agent decision-making, roadmap updating, clustering and pruning. We can first observe that the reward estimation time dominates the overall time consumption across all dataset sizes. This phenomenon can be primarily attributed to the computationally intensive nature of the downstream tasks evaluation process. In addition, the time cost of reward estimation increases proportionally with the size of the dataset, resulting in a linear scalability of TCTO in terms of time complexity. In summary, the main temporal bottleneck of this framework, as well as other iterative-feedback approaches, is the downstream task evaluation in the reward estimation component.

#### A.3.2 SPACE COMPLEXITY ANALYSIS

Table 3: The space complexity analysis

This experiment aims to answer the question: *Does* TCTO *have a good spatial scalability?* Table 3 presents the space complexity of each agent and the total number of parameters in TCTO. Since our model's reinforcement learning structure remains fixed

| Hidden dim. | Output dim. | Embedding dim. | Head agent param. | Operation agent param. | Operand agent param. | Total param. |
|---|---|---|---|---|---|---|
| 32 | 64 | 7 | 53993 | 14516 | 20213 | 177444 |
| 32 | 32 | 7 | 29129 | 8116 | 10613 | 95716 |
| 32 | 64 | 16 | 53993 | 14516 | 21257 | 179532 |
| 64 | 32 | 7 | 51625 | 8116 | 10613 | 140708 |
| 64 | 64 | 7 | 94921 | 14516 | 20213 | 259300 |
| 64 | 64 | 16 | 94921 | 14516 | 21257 | 261388 |

and decoupled with dataset size, it maintains constant space complexity even when exploring large-scale datasets. We configure various dimensions for RGCN hidden layers, output layers, and operation embeddings to assess their impact on space complexity. We can observe that the scale of the head cluster agent correlates with the dimensions of RGCN hidden and output layers, as it encodes the roadmap during the initial step. Similarly, the parameter scale of the operation agent is influenced by the dimension of RGCN output layers, as this agent makes decisions based on state information derived from roadmap embeddings. The operand cluster agent's space complexity is higher due to its inclusion of an additional embedding layer for encoding mathematical operations within the value network. Notably, our model employs a dual value-network structure within the deep Q-Learning framework, resulting in a total parameter count twice the sum of the parameters of the three cascading agents.

Table 4: Robustness check of TCTO with distinct ML models on different datasets

| | RFR | Lasso | XGBR | SVM-R | Ridge-R | DT-R | MLP |
|---|---|---|---|---|---|---|---|
| ATF | 0.433 | 0.277 | 0.347 | 0.276 | 0.187 | 0.161 | 0.197 |
| ERG | 0.412 | 0.162 | 0.331 | 0.278 | 0.256 | 0.257 | 0.300 |
| NFS | 0.434 | 0.169 | 0.391 | 0.324 | 0.261 | 0.293 | 0.306 |
| RDG | 0.434 | 0.193 | 0.299 | 0.287 | 0.218 | 0.257 | 0.279 |
| TTG | 0.424 | 0.163 | 0.370 | 0.329 | 0.261 | 0.294 | 0.308 |
| GRFG | 0.451 | 0.185 | 0.435 | 0.363 | 0.265 | 0.197 | 0.208 |
| **TCTO** | **0.495** | **0.370** | **0.444** | **0.384** | **0.317** | **0.350** | **0.310** |

(a) Housing Boston

| | RFC | XGBC | LR | SVM-C | Ridge-C | DT-C | KNB |
|---|---|---|---|---|---|---|---|
| ATF | 0.669 | 0.608 | 0.634 | 0.664 | 0.633 | 0.564 | 0.530 |
| ERG | 0.683 | 0.703 | 0.659 | 0.571 | 0.654 | 0.580 | 0.537 |
| NFS | 0.659 | 0.607 | 0.627 | 0.676 | 0.646 | 0.613 | 0.577 |
| RDG | 0.627 | 0.607 | 0.623 | 0.669 | 0.660 | 0.609 | 0.577 |
| TTG | 0.650 | 0.607 | 0.633 | 0.676 | 0.646 | 0.599 | 0.577 |
| GRFG | 0.692 | 0.648 | 0.642 | 0.486 | 0.663 | 0.580 | 0.552 |
| **TCTO** | **0.742** | **0.730** | **0.706** | **0.701** | **0.689** | **0.652** | **0.587** |

(b) Messidor_features

Table 5: A case study on the ten most significant features of both original and transformed datasets for Housing Boston and White Wine Quality

| Housing Boston | | TCTO$^{-g}$ | | TCTO | |
|---|---|---|---|---|---|
| feature | importance | feature | importance | feature | importance |
| lstat | 0.362 | quan_trans(lstat) | 0.144 | $v_{18} : \sqrt{\|v_{17}\|}$ | 0.080 |
| rm | 0.276 | lstat | 0.135 | $sta(v_{17})$ | 0.077 |
| dis | 0.167 | quan_trans(rm) | 0.126 | $sta(\sqrt{\|v_{17}\|})$ | 0.054 |
| crim | 0.072 | rm | 0.119 | $sta(v_{16})$ | 0.054 |
| rad | 0.032 | (dis+(...))-quan(lstat) | 0.076 | $sta(\sqrt{\sqrt{v_{18}}})$ | 0.053 |
| black | 0.032 | (dis*(...))+(...)+(dis+...) | 0.050 | $v_{16} : \frac{1}{\sin v_{12} - v_0}$ | 0.053 |
| age | 0.030 | (dis+...)+(...)-(zn+(...)) | 0.048 | $sta(v_{24})$ | 0.050 |
| nox | 0.011 | (dis+...)-(...)+quan(rm) | 0.028 | $\min(v_5)$ | 0.044 |
| ptratio | 0.007 | (dis+..lstat)-(...+rad) | 0.016 | $v_{17} : \sqrt{\|v_{16}\|}$ | 0.037 |
| indus | 0.005 | (dis+..crim)-(...+rad) | 0.015 | $v_{12}$ | 0.025 |
| 1-RAE:0.414 | Sum:0.993 | 1-RAE:0.474 | Sum:0.757 | 1-RAE:0.494 | Sum:0.527 |

| Wine Quality White | | TCTO$^{-g}$ | | TCTO | |
|---|---|---|---|---|---|
| feature | importance | feature | importance | feature | importance |
| alcohol | 0.118 | quan_trans(alcohol) | 0.043 | $v_2 + v_{30}$ | 0.026 |
| density | 0.104 | alcohol | 0.036 | $\sin(\sin(f_0)) + v_{30}$ | 0.025 |
| volatile | 0.099 | ((den...)+(alc...)/(...)) | 0.028 | $v_5 + v_{30}$ | 0.024 |
| free sulfur | 0.093 | quan_trans(density) | 0.028 | $\sin(f_0) + v_{30}$ | 0.023 |
| total sulfur | 0.092 | density | 0.028 | $v_2$ | 0.023 |
| chlorides | 0.091 | (den/(...))+(dens...)/(...) | 0.026 | $v_3 + v_{30}$ | 0.023 |
| residual | 0.087 | (den/(...))+((...)/tan(...)) | 0.024 | $v_6 + v_{30}$ | 0.021 |
| pH | 0.082 | (den/...)-(...+stand(...)) | 0.023 | $v_7 + v_{30}$ | 0.021 |
| citric acid | 0.081 | (citr/(...)+(...)/tanh(...)) | 0.023 | $v_0 + v_{30}$ | 0.021 |
| fixed acidity | 0.078 | (free/(...)+(...)/tanh(...)) | 0.023 | $v_{11} + v_{30}$ | 0.021 |
| F1-score:0.536 | Sum:0.924 | F1-score:0.543 | Sum:0.282 | F1-score:0.559 | Sum:0.228 |

### A.3.3 ROBUSTNESS CHECK

This experiment aims to answer the question: *Are our generative features robust across different machine learning models used in downstream tasks?* We evaluate the robustness of the generated features on several downstream models. For regression tasks, we substitute the Random Forest Regressor (RFR) with Lasso, XGBoost Regressor (XGB), SVM Regressor (SVM-R), Ridge Regressor (Ridge-R), Decision Tree Regressor (DT-R), and Multilayer Perceptron (MLP). For classification tasks, we assess the robustness using Random Forest Classifier (RFC), XGBoost Classifier (XGB), Logistic Regression (LR), SVM Classifier (SVM-C), Ridge Classifier (Ridge-C), Decision Tree Classifier (DT-C), and K-Neighbors Classifier (KNB-C). Table 4 presents the results in terms of 1-RAE for the Housing Boston dataset and F1-score for the Messidor_features dataset, respectively. We can observe that the transformed features generated by our model consistently achieved the highest performance in regression and classification tasks among each downstream machine learning method. The underlying reason is that these features contain significant information that is capable of fitting into various machine learning tasks. Therefore, this experiment validates the effectiveness of our model in generating informative and robust features for various downstream models.

### A.3.4 CASE STUDY ON GENERATED FEATURES

This experiment aims to answer the question: *Can our model reuse the high-value sub-transformation and generate a high-quality feature space?* Table 5 presents the Top-10 most important features generated by the original dataset, our proposed method, and its feature-centric variants (i.e., TCTO$^{-g}$). We can first observe that TCTO has reused many high-value sub-transformations, such as node $v_{17}$ in Housing Boston and node $v_{30}$ in Wine Quality White. Compared to TCTO$^{-g}$, the roadmap-based model tends to reuse important intermediate nodes, transforming them to generate more significant features. A possible reason for this is that our model effectively utilizes historical information from the roadmap, identifying optimal substructures and exploring and transforming these crucial nodes, thereby utilizing the historical sub-transformations. Another point to note is that the transformed feature's importance score in our model tends to be more balanced compared to the original dataset and its variant, e.g., the sum of the top-10 feature importance is lower. Since our model has better performance, we speculate that our framework comprehends the properties of the feature set and ML models to produce numerous significant features by combining the original features. Regarding the record of feature transformations shown in Table 5, which is depicted through a formula combining both original and intermediate features, full traceability is also achieved. Such characteristics of traceability might help experts find new domain mechanisms.

### A.3.5 ANALYSIS ON THE WEIGHT OF REWARD FUNCTION

This experiment aims to answer the question: *How does the trade-off between performance and complexity impact the performance?* We conducted a preliminary experiment using the Airfoil dataset to analyze the impact of varying reward weights during the optimization stage. Figure 12 shows that only the complexity reward or the performance reward is used exclusively, the performance is noticeably lower. This result suggests that while performance rewards encourage the agent to generate high-value features, overly complex features can be detrimental to the downstream task. With a balanced weight of them, the performance fluctuates slightly. Based on these preliminary results, we concluded that a ratio of 1:1 between feature quality and complexity, providing stable and reliable performance.

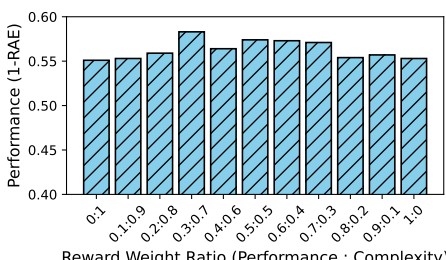

Figure 12: Impact of varying weights between performance and complexity rewards

### A.3.6 ANALYSIS ON THE SCALABILITY ON LARGE-SCALE DATASETS

This experiment aims to answer the question: *Can our approach scale to large-scale datasets?* We categorize large-scale datasets into two types: large-sample and high-dimensional datasets. As shown in Section A.3.1, the main temporal bottleneck of TCTO lies in the downstream task evalua-

Table 6: Comparison of Baseline Methods and TCTO on ALBERT

| Dataset | Original | RDG | ERG | LDA | AFAT | NFS | TTG | GRFG | DIFER | FETCH | OpenFE | TCTO |
|---|---|---|---|---|---|---|---|---|---|---|---|---|
| ALBERT | 0.674 | 0.678 | 0.619 | 0.530 | * | 0.680 | 0.681 | * | * | * | 0.679 | 0.681 |
| Newsgroups | 0.568 | 0.556 | 0.545 | 0.230 | 0.544 | 0.553 | 0.546 | * | * | * | 0.544 | 0.576 |

We report F1-score for ALBERT and Macro-F1 for Newsgroups.
'*' indicates that the method ran out of memory or took too long.

tion within the reward estimation component.

**Scalability with large-sample datasets:** ALBERT is a large-sample dataset with 425,240 samples and 78 features. The time required for the downstream task is approximately 16 minutes per step which is unacceptable. To address this issue, we switched to a more efficient downstream model LightGBM, which offers faster speed. Table 6 demonstrates that TCTO can effectively scale by leveraging more efficient models for large-sample datasets. We observed that all methods exhibit limited gains, and the results suggest that feature transformation methods have limited impact on extremely large-sample datasets, consistent with existing work (see Table 3 in study (Zhang et al., 2023)). The key reason for this is that neural networks can learn latent patterns from sufficient samples, making additional feature transformation less essential.

**Scalability with high-dimensional datasets:** Newsgroups is a high-dimensional dataset with 13,142 samples and 61,188 features. The time required for the downstream task is approximately 5 minutes per step. For high-dimensional datasets, we employed a pruning strategy to remove irrelevant root nodes before exploring the dataset. Table 6 shows that TCTO outperforms baseline methods in terms of Macro-F1. The pruning strategy helps mitigate the complexity of high-dimensional datasets, speeding up the process while maintaining performance.

In conclusion, our experiments demonstrate that TCTO is scalable and performs well on both large-sample and high-dimensional datasets when appropriate strategies are employed.

## A.4 DISCUSSION ON FUTURE WORK

In this section, we outline three potential future milestones: enhancing the framework, incorporating large language models, and exploring further application scenarios.

### A.4.1 IMPROVEMENT ON THE FRAMEWORK

While TCTO demonstrate promising advances toward building a roadmap for automated feature transformation, our analysis has identified some primary bottlenecks in our approach:

**Regarding the Framework of Iterative-Feedback Approaches.** Those approaches result in a time-consuming nature of the evaluations on downstream tasks. Toward this metric, the reward feedback will be more precise and directed. However, this phase often requires extensive computational resources and time Ying et al. (2024), especially when dealing with large datasets and complex models. In future work, we aim to balance the trade-off between efficacy and efficiency by adaptively integrating some unsupervised evaluation metrics into the iterative-feedback framework, thus making it more suitable for huge datasets.

**Regarding the Limitation of State Representation Method:** The state representation method could be treated as the 'eye' of each cascading agent. In our study, the state representation method is consisted of statistical information of (generated) features and the historical feature-feature crossing maintained by the roadmap. Although applying graph modeling technique on the evolving could capture the latent information within the feature-feature crossing, the discussion of feature-level state representation is still limited. Inspired by the study Xiao et al. (2024), our work will enhance the state representation technique by integrating advanced deep learning methods, including AutoEncoder Kingma (2013), Graph Attention Network Veličković et al. (2017), and large language models in the future.

**Regarding the Limitation of Applications on Large-scale Datasets:** While TCTO and feature transformation methods show promising results for small-scale datasets, further research is needed to improve their adaptability to large-scale and high-dimensional datasets. Specifically, future work will focus on the following areas: *Optimizing Feature Transformation Methods for Large-sample Datasets*: We aim to develop more scalable feature transformation methods that can better handle large-scale datasets without introducing significant computational bottlenecks. This may involve in-

corporating more efficient algorithms for feature generation and selection. *Enhancing Feature Pruning Techniques for High-dimensional Datasets*: Given the challenges posed by high-dimensional datasets, we plan to investigate advanced feature pruning strategies that can more effectively identify and retain the most relevant features while minimizing performance loss. Additionally, exploring hybrid approaches combining feature selection and transformation could enhance efficiency.

### A.4.2 LARGE LANGUAGE MODELS AS DATA SCIENTISTS

Recently some research has utilized Large Language Model(LLM) to generate high-quality features (Hassan et al., 2023; Long et al., 2024; Zhang et al., 2024; Hollmann et al., 2024). However, LLMs exhibit limitations from two perspectives: the semantic understanding of feature names and the issue of hallucination.

**Regarding the Limitation of Feature Names Comprehension**: LLM-based methods utilize the comprehensive ability, conducting transformation to generate high-quality features based on the semantic of feature names. However, due to data quality issues, feature names are frequently anonymous or missing in some case, such as federated learning scenarios. With semantic blinding, Hollmann et al. (2024) found *a strong drop in performance* based on their experimental conclusion.

**Regarding the Issue of Hallucination**: When LLMs are engaged in certain feature engineering tasks, such as evaluating feature importance, they often encounter the hallucination problem. The hallucination problem in LLMs can result in *the generation of irrelevant or unsupported content* (Hassan et al., 2023). Due to hallucination problems, Jeong et al. (2024) reported that without providing dataset, LLMs can produce a precise numerical value for importance.

In contrast to these approaches, TCTO operates independently of the semantic context of feature names, allowing for broader application scenarios. We also acknowledge that integrating common or injected knowledge in LLMs has the potential to greatly improve the scalability, performance, and efficiency of our framework. It also worth noting that Jeong et al. (2024), GPT-4 (et al., 2024) exhibits notable hallucination problems. However, we have noticed a significant improvement following the release of GPT-4o. We provide a preliminary experiment in the following figure, (regarding omitted prompts, please refer to (Jeong et al., 2024)), where LLMs demonstrate a reasonable capacity to evaluate the importance of features. In future work, we will explore how LLMs can be utilized to comprehend and evaluate the roadmap that identified by TCTO, focusing on research aspects such as node clustering, reward feedback and node pruning.

---

**Preliminary Experiment on GPT-4o: Feature Importance Notion**

**System:** (Omitted) # Prompts are omitted.
**User:** Provide a score and reasoning for "Number of times pregnant" formatted according to the output schema above:
**GPT-4o:** { "reasoning":"The number of times a patient has been pregnant is associated with gestational diabetes and overall metabolic health. Multiple pregnancies can increase the risk of developing diabetes later in life, making this feature significant for prediction.", "score": 0.75 } # GPT-4 can also support this analysis.
**User:** Is there a specific notion of feature importance that is captured in your score?
**GPT-4o:** Yes, the score reflects the degree to which the feature contributes to the model's ability to predict the target variable—in this case, the likelihood of having diabetes. It incorporates both the statistical relevance of the feature in relation to diabetes risk and its clinical significance based on existing medical knowledge. This can involve techniques like correlation analysis, feature selection methods, or model-specific importance metrics, ensuring the score represents a blend of empirical data and domain expertise. # GPT-4 can also support this analysis, but hallucination could happened.
**User:** What kind of statistical analysis are you referring to? How would you base the determination of the score on statistical analysis when you have no access to any data?
**GPT-4o:** In a real scenario, statistical analysis methods like the following could be employed to determine feature importance scores based on available data: Correlation Analysis: Examining the correlation coefficients (e.g., Pearson or Spearman) between each feature and the target variable to identify how strongly each feature relates to the likelihood of diabetes.
*Feature Selection Techniques:* Recursive Feature Elimination (RFE): Iteratively removing fea-

---

tures and evaluating model performance to identify which features contribute most to predictive power. Lasso Regression: Using regularization to shrink coefficients of less important features to zero, allowing identification of important features.

*Tree-based Methods:* Algorithms like Random Forest or Gradient Boosting can provide feature importance scores based on how much each feature contributes to reducing uncertainty in predictions.

*Statistical Tests:* Using tests such as the Chi-square test for categorical features or t-tests for continuous features to assess their significance in relation to the target variable. Without access to data, the scores I provided were based on common clinical knowledge and established relationships in the medical literature regarding diabetes risk factors. The scores are hypothetical and illustrative, meant to guide the conceptual understanding of how feature importance might be assessed in practice. # GPT-4 can not provide reason on its previous analysis and will response SORRY.

### A.4.3 APPLICATION SCENARIOS OF FEATURE GENERATION

For years, feature engineering has been a significant step Sambasivan et al. (2021); Strickland (2022) before applying computational statistics or machine learning methods to data, and its importance has grown with the growing trend of interdisciplinary research between artificial intelligence and various fields, such as life sciences (Ofer & Linial, 2015; Chen et al., 2020; Chicco et al., 2022; Bonidia et al., 2022) and material sciences (Dai et al., 2020; Kaundinya et al., 2021; Xiang et al., 2021; Anand et al., 2022). Given the complexity inherent in scientific data, automated feature transformation methods like the one proposed in this work have significant potential to advance various AI4Science disciplines. In bio-informatics and computational biology, our approach can aid in extracting pivotal gene, protein, or metabolites combinations (as depicted in our motivation, Figure 1) from high-throughput sequencing data, enhancing the identification of gene networks associated with diseases. In the realm of chemistry and drug discovery, chemically meaningful features can automatically be generated to improve the accuracy of molecular activity and toxicity predictions, thereby accelerating the development of new pharmaceuticals. In future research, we plan to advance our study by assisting life sciences experts in identifying combinations within population cohort data. The feature transformation roadmap can ensure traceability and interpretability, thereby aiding scientific discovery and enhancing the model's accuracy in early disease detection.

