# OpenReview forum: "Build Roadmap for Automated Feature Transformation: A Graph-based Reinforcement Learning Approach"
_ICLR.cc/2025/Conference — Submitted to ICLR 2025_

### Official Review · Reviewer_TgFr · 2024-10-24

**Soundness:** 2
**Presentation:** 2
**Contribution:** 3
**Rating:** 6
**Confidence:** 2

**Summary:**

The paper deals with the automated generation of features. The generation process consists of several steps, which are represented as a graph. The graphs are to be optimized using multi-agent reinforcement learning.

**Strengths:**

* It can be seen (among other things from the large number of specific illustrations) that a lot of effort was put into preparing the paper

**Weaknesses:**

* I find the text very badly written. Examples follow. The novelty and benefits of the method are hard for me to understand.
* It seems to me that there is too much material for a conference paper, the number of pages is simply not enough to present it in a convincing way.

Details, examples and further comments:
* I don't think “roadmap” is a suitable term, “schedule” or "sequence" would probably be better.
* The title sounds strange. Wouldn't "Optimization of transformation sequences for automated feature generation“ be better?
* The abstract uses terms that are incomprehensible:\
mathematical feature-feature crossing\
the roadmap of feature transformation
* „Feature transformation task aims to generate high-value features and improve the performance of downstream machine learning tasks using the mathematical feature-feature crossing” needs to be reformulated.
* "Classic machine learning is highly dependent on the structure of the model, the activation function" cannot be said in this way, it seems to refer exclusively to neural networks and not to classical machine learning in general.
* A reference should be given for "a cascading multi-agent reinforcement learning (MARL) algorithm", because it is not generally known what “cascading multi-agent reinforcement learning” is.
* “we present an innovative framework” -> “we present a novel framework”
* In the loss function, Equation 8, the square is probably missing.
* "In this study, we introduce TCTO, an automated feature transformation framework. Our method emphasizes a transformation-centric approach, in which a transformation roadmap is utilized to systematically track and manage feature modifications." should be reworded. What is the information content? What should be expressed?
* I think that the Abstract and Conclusion need to be completely rewritten.

**Questions:**

* How were the small uncertainties in Table 1 achieved? How often were the experiments repeated?

---

> ### Author Response · Authors · 2024-11-19
> **Discussion about your concerns**
>
> Dear Reviewer TgFr,\
> Thank you for your valuable comments and feedback on our submission.
> We sincerely appreciate the time and effort you have invested in reviewing our work!
> We are committed to addressing your concerns and will revise the manuscript accordingly.
>
> (1) **Response to your comments on details 1 & 2:**\
> We acknowledge your concerns regarding the feature transformation process in existing methods[1].
> In these methods, the feature transformation is modeled as a sequence generation process; however, the candidate feature set is constrained to the set of features available at the current transformation step (see Section Our Perspective and Contribution (1)).
> This approach limits the flexibility of the transformation process. More importantly, past transformations' latent correlations and mathematical characteristics are not captured in the sequence-based approach, and the historical insights from previous feature transformations are discarded (see Section Our Perspective and Contribution (2)).
> Consequently, this lack of transformation agility reduces the expressiveness of the model in capturing the correlations among features.
>
> In contrast, our approach utilizes a roadmap, which is a traceable transformation graph that retains all transformation steps and their interconnections.
> This roadmap allows the model to not only understand the current state of features but also access historical transformations and their relationships.
> This is why we prefer the term "roadmap" rather than "schedule" or "sequence."
> An example of this roadmap is illustrated in Figure 2.
> Based on these considerations, we believe that the title "BUILD ROADMAP FOR AUTOMATED FEATURE TRANSFORMATION: A GRAPH-BASED REINFORCEMENT LEARNING APPROACH" better reflects the core idea of our work, emphasizing both the structure of our data and the optimization methods we employ.
>
> [1] Xiao M, Wang D, Wu M, et al. Traceable automatic feature transformation via cascading actor-critic agents[C]//Proceedings of the 2023 SIAM International Conference on Data Mining (SDM). Society for Industrial and Applied Mathematics, 2023: 775-783.
>
> (2) **Response to your comments on details 3 & 10:**\
> The term "mathematical feature-feature crossing" refers to a mathematical operation performed between two features to generate a new one (e.g., $BMI = weight / height^2$, as shown in Figure 1).
> The term "roadmap of feature transformation" refers to a graph that encapsulates the entire transformation process (see Figure 2 in Section "Preliminary").
> We acknowledge that the abstract could benefit from clearer wording.
> We will revise the abstract and the layout of the figures in the introduction to enhance clarity and ensure that these concepts are expressed more intuitively.
>
> (3) **Response to your comments on details 4, 5, & 9:**\
> Thank you for pointing out these confusions!\
> 4: "Feature transformation tasks aim to generate high-value features through mathematical feature-feature crossing, which can enhance the performance of downstream machine learning tasks."\
> 5: "Classic machine learning is highly dependent not only on the structure of the model but also on the quality of the training data."\
> 9: "We present TCTO, an automated framework for feature transformation. Our approach focuses on managing feature modifications through a transformation roadmap, which systematically tracks and organizes the transformation process to ensure optimal feature generation."\
>
> (4) **Response to your comment on detail 6:**\
> We introduced the concept of cascading agents in Section 3.3. To further clarify, we will cite relevant works on cascading multi-agent reinforcement learning in the revised manuscript to provide additional context. \
> [2] Busoniu L, Babuska R, De Schutter B. A comprehensive survey of multiagent reinforcement learning[J]. IEEE Transactions on Systems, Man, and Cybernetics, Part C (Applications and Reviews), 2008, 38(2): 156-172.\
> [3] Panait L, Luke S. Cooperative multi-agent learning: The state of the art[J]. Autonomous agents and multi-agent systems, 2005, 11: 387-434.
>
> (5) **Response to your comment on detail 8:**\
> We apologize for the confusion regarding the loss function in Equation (8). As stated in line 375, the parameters of the prediction network are updated through gradient descent to minimize the loss. We will revise Equation (8) to address the issue you pointed out.
>
> (6) **Response to your question 1:**\
> The small uncertainties in Table 1 were achieved through 5 times repeated experiments. We will clarify this process in the experiment setting.
>
> We sincerely appreciate your detailed and constructive feedback, which has been invaluable in helping us improve our manuscript.
> If you find that our revisions satisfactorily address your concerns, we kindly ask you to consider increasing the score of our submission.
> Thank you for your time and thoughtful review! If you have any further questions or suggestions, please feel free to discuss with us.

---

> > ### Author Response · Authors · 2024-12-01
> >
> > Dear reviewer TgFr,
> >
> > We want to express our sincere gratitude for revising your score on our paper. As the discussion period approaches its conclusion, we would like to address any remaining concerns you may have regarding your marginally negative score.
> >
> > We appreciate your feedback and look forward to your response.
> >
> > Best regards,\
> > The Authors.

---

### Official Review · Reviewer_nnCL · 2024-11-03

**Soundness:** 3
**Presentation:** 3
**Contribution:** 3
**Rating:** 6
**Confidence:** 2

**Summary:**

In this paper, the authors present TCTO, a graph-based reinforcement learning framework designed for automated feature transformation. The approach addresses limitations in current methods, such as the lack of historical insight utilization and insufficient flexibility in transformation exploration. By constructing a transformation roadmap with nodes representing feature states, TCTO leverages a cascading multi-agent system to dynamically select transformations, reuse effective paths, and prune inefficient ones. The experimental results demonstrate that TCTO outperforms existing methods in generating high-quality features, suggesting its potential to enhance feature engineering in machine learning tasks.

**Strengths:**

This paper has several notable strengths. Firstly, the authors present a well-motivated framework that addresses clear gaps in current automated feature transformation methods, such as the need for effective historical data utilization and robust backtracking. The proposed TCTO framework is innovative in its use of a graph-based roadmap and cascading multi-agent reinforcement learning, which enhance the flexibility and adaptability of the transformation process. Additionally, the authors provide a comprehensive experimental evaluation across diverse datasets, which convincingly demonstrates TCTO’s superior performance compared to traditional methods. This solid empirical foundation supports the framework's potential for broad applicability in feature engineering for machine learning tasks.

**Weaknesses:**

While this paper offers a promising framework, it has some weaknesses. Firstly, the explanation of the cascading multi-agent system and its decision-making processes could benefit from more clarity and detail, as the current description may be challenging for readers to fully grasp without additional context. Additionally, the computational complexity of TCTO is not thoroughly analyzed, especially regarding scalability to larger datasets, which may impact its practical applicability. Finally, while the experimental results are extensive, the paper could further strengthen its claims by providing more insight into specific scenarios or datasets where TCTO may struggle, thereby clarifying the framework’s limitations and potential areas for improvement.

**Questions:**

1.How does the computational complexity of TCTO scale with larger datasets, and are there any strategies to mitigate potential performance bottlenecks?
2.Are there scenarios or specific types of datasets where TCTO’s performance may be limited, and if so, what adjustments might be necessary to enhance its adaptability?

---

> ### Author Response · Authors · 2024-11-19
> **(1/2) Discussion about computational complexity and scalability on large-scale datasets**
>
> Thank you for the time and effort you have dedicated to reviewing our paper.
> We greatly appreciate your insightful comments and recognize the efforts and contributions of our work.
> The following are our detailed responses to your weaknesses and questions:
>
> **Re: weakness 2 and question 1**\
> Thank you for your insightful comments. We recognize the importance of addressing the computational complexity and scalability of TCTO for larger datasets.
>
> In Section A.3.1, we analyzed time consumption and identified that the primary bottleneck arises from the downstream task. In our experiments, we used Random Forest (RF) implemented in scikit-learn. But as the dataset size increases, RF requires significantly more time. Table 1 shows the time consumption for two large-scale datasets.
>
> Table 1. Time consumption on Large-Scale Datasets (RF Model)
>
> |Dataset|#Samples|#Features|Time Consumption|
> |--|--|--|--|
> |ALBERT|425,240|78|~16 mins|
> |newsgroups|13,142|61,188|~5 mins|
>
> **TCTO can scale to large-sample datasets:**\
> The time consumption for ALBERT (in Table 1) indicates that such extensive durations are unacceptable. To address this, we switched to a more efficient downstream model, LightGBM. The underlying insight is that performance on the downstream model only serves as a reward signal for the cascading agents. Table 2 compares the time consumption and performance of baselines and TCTO with LightGBM as the downstream model.\
> Table 2: Comparison of Baseline Methods and TCTO on ALBERT (LightGBM Model)
>
> |ALBERT|Original|RDG|ERG|LDA|AFAT|NFS|TTG|GRFG|DIFER|TCTO|
> |--|--|--|--|--|--|--|--|--|--|--|
> |F1-Score|0.674|0.678|0.619|0.530|*|0.680|0.679|*|*|0.681|
> |Per Step Time (s)|3.41^|87.40|7.53|2046.7^|*|28.17|49.83|*|*|8.51|
>
> Note: * indicates methods that ran out of memory or took too long. ^ indicates total time consumption.
>
> **All feature transformation methods show limited improvement on large-sample datasets:**\
> These results demonstrate that TCTO can effectively scale to large datasets by leveraging more efficient downstream models, such as LightGBM. However, as shown in Table 2, TCTO shows limited improvement on large-sample datasets. We observe that all methods exhibit limited gains on such datasets. With a sufficient number of samples, machine learning models are able to fit the data well, and the additional information generated by feature transformation becomes less essential. In contrast, feature transformation methods are particularly beneficial for small-scale datasets (see dataset description in [1][2]). These findings align with prior research, which shows that feature transformation has limited impact on extremely large datasets (see Table 3 in [3]).
>
> [1] Li L, Wang H, Zha L, et al. Learning a data-driven policy network for pre-training automated feature engineering[C]//The Eleventh International Conference on Learning Representations. 2023.\
> [2] Wang D, Xiao M, Wu M, et al. Reinforcement-enhanced autoregressive feature transformation: Gradient-steered search in continuous space for postfix expressions[J]. Advances in Neural Information Processing Systems, 2023, 36: 43563-43578.\
> [3] Zhang T, Zhang Z A, Fan Z, et al. OpenFE: automated feature generation with expert-level performance[C]//International Conference on Machine Learning. PMLR, 2023: 41880-41901.
>
> **TCTO can scale to high-dimensional datasets:**\
> High-dimensional datasets often contain some unimportant features that may hinder the feature transformation. We performed experiments on high-dimensional Newsgroups dataset, which contains 13,142 samples and 61,188 features. As shown in Table 1, the evaluation time was approximately 5 minutes per run. We employed pruning strategy to remove irrelevant root nodes before exploring the dataset. The performance and time consumption of TCTO with LightGBM as the downstream model, are shown in Table 3.\
> Table 3: Comparison of Baseline Methods and TCTO on Newsgroups
>
> |Newsgroups|Original|RDG|ERG|LDA|AFAT|NFS|TTG|GRFG|DIFER|TCTO|
> |--|--|--|--|--|--|--|--|--|--|--|
> |Macro F1|0.568|0.556|0.545|0.230|0.544|0.553|0.546|*|*|0.576|
> |Per Step Time (s)|15.44^|4.37|347.0|12.24^|82.20^|21.23|19.92|*|*|18.02|
>
> Note: * indicates that the method ran out of memory or took too long. ^ indicates total time consumption.
>
> **Pruning strategy can filter out unimportant nodes:**\
> These results demonstrate that TCTO outperforms baseline methods in terms of Marco-F1 score. The pruning strategy helps mitigate the complexity of high-dimensional datasets and speeds up the process while maintaining performance.
>
> In conclusion, TCTO can scale to large-sample and high-dimensional datasets by employing efficient downstream models and implementing a pruning strategy, respectively. With large-sample datasets, feature transformation has limited performance improvement. With high-dimensional datasets, TCTO prunes original nodes to mitigate the complexity of datasets and speeds up feature transformation process.

---

> ### Author Response · Authors · 2024-11-19
> **(2/2) Discussion about adaptability, limitations and clarity of expression**
>
> **Re: weakness 3 and question 2**\
> Thank you for your valuable feedback. We understand your concern regarding the adaptability and limitations of TCTO. As discussed in Appendix 4.1, TCTO has certain limitations that may affect its performance under specific conditions. We will expand on these limitations to provide a more comprehensive understanding of the framework's boundaries.
>
> Apart for we have discussed, we will add this disscussion in the final version:\
> In small-scale datasets, where the number of samples is limited and insufficient for machine learning models to learn complex patterns, feature transformation can significantly improve model performance.  By generating high-value features that better capture underlying data patterns, feature transformation methods provide additional context, making it easier for models to extract meaningful insights from the available data. This is particularly important in privacy-sensitive domains, such as medical and financial datasets, where data may be constrained in both sample size and feature space. In these cases, feature transformation can serve as an effective tool for uncovering latent knowledge. However, in large-sample datasets, machine learning models often fit the data well, reducing the need for additional information from feature transformation. As a result, the performance improvements from transformation methods are less pronounced. In high-dimensional datasets, the presence of irrelevant features can increase computational time and degrade model performance. While feature pruning techniques can reduce dimensionality, they may also lead to performance degradation if important features are removed.
>
> Future Work:
> While TCTO and feature transformation methods show promising results for small-scale datasets, further research is needed to improve their adaptability to large-scale and high-dimensional datasets. Specifically, future work will focus on the following areas:\
> **Optimizing Feature Transformation for Large Datasets:** We aim to develop more scalable feature transformation methods that can better handle large-scale datasets without introducing significant computational bottlenecks. This may involve incorporating more efficient algorithms for feature generation and selection. **Enhancing Feature Pruning Techniques:** Given the challenges posed by high-dimensional datasets, we plan to investigate advanced feature pruning strategies that can more effectively identify and retain the most relevant features while minimizing performance loss. Additionally, exploring hybrid approaches combining feature selection and transformation could enhance efficiency.
>
> **Re: weakness 1** \
> We reorganized and rewrite the content of cascading agents to impove the clarity, specifically:\
> (Line 196 - Line 200) Multi-agent Reinforcement Learning-based Transformation Decision:
> Reinforcement learning has proven effective in addressing complex decision-making challenges across various domains. We employ three cascading agents that collaboratively construct unary and binary mathematical transformations. These agents operate sequentially to select the optimal head cluster, mathematical operation, and operand cluster, respectively. The chosen features undergo the specified mathematical operations, resulting in the generation of new features and the creation of new nodes within the roadmap. Additional details regarding the decision-making process will be provided in Section 3.3, Cascading Reinforcement Learning Agents.
>
> (Line 313 - Line 315) Cascading Reinforcement Learning Agents: Figure 7 shows an example of the cascading agents' decision-making process. We utilize a series of cascading agents, each performing a specific task in sequential order. These agents collaborate in a step-by-step decision-making process, where the output of one agent serves as the input for the next. The first agent (head cluster agent) is responsible for selecting the head cluster, the second (operation agent) for choosing the most appropriate mathematical operation, and the third (operand cluster agent) for identifying the operand cluster. By using this cascading structure, each decision is informed by the context set by the previous agents, leading to a more efficient decision-making process. The details of each agent are as follows:
>
> Thank you once again for your time and effort. We hope that the responses provided address your concerns and sincerely hope you will reconsider the rating. If you have any further questions or require additional clarification, please do not hesitate to discuss with us.

---

> > ### Comment · Area_Chair_CLEW · 2024-11-24
> > **Please respond to rebuttal ASAP**
> >
> > Dear reviewer,
> > The process only works if we engage in discussion. Can you please respond to the rebuttal provided by the authors ASAP?

---

> ### Author Response · Authors · 2024-12-01
>
> Dear reviewer nnCL,
>
> We sincerely appreciate your revision of the score. As the discussion period draws to a close, we hope to address any remaining concerns you may have and further adjust your score accordingly.
> We appreciate your feedback and look forward to your response.
>
> Best regards,\
> The Authors.

---

### Official Review · Reviewer_cE2X · 2024-11-05

**Soundness:** 3
**Presentation:** 3
**Contribution:** 2
**Rating:** 5
**Confidence:** 2

**Summary:**

This paper introduces an automated feature transformation framework designed to enhance downstream machine learning model performance. The TCTO framework leverages a reinforcement learning-based graph structure to maintain a roadmap of feature transformations, enabling efficient exploration and backtracking of transformation pathways. TCTO uses a multi-agent reinforcement learning approach, clustering and encoding transformation states to strategically apply feature transformations. Experiments on multiple datasets demonstrate TCTO's performance over existing methods by improving robustness and flexibility in feature generation.

**Strengths:**

1. While mostly clear, certain sections (e.g., cascading agent decision process) could benefit from additional details.

2. The framework is well-supported by experimental evidence showing its adaptability across different datasets and improvement in downstream model performance.

3. TCTO introduces a novel approach to automated feature engineering by employing a transformation-centric methodology with a graph-based roadmap, overcoming limitations of existing feature transformation methods.

4. The approach’s ability to backtrack and optimize feature transformations dynamically makes it highly applicable in real-world ML tasks where feature diversity and stability are crucial.

**Weaknesses:**

1. While effective on a range of datasets, it is unclear how well TCTO scales with extremely high-dimensional data or very large datasets, as the pruning strategy may require fine-tuning in these cases.

2. The cascading decision-making process is intricate, and further simplification or additional visuals might aid understanding.

3. The reward structure combines performance and complexity, but further discussion on how these metrics are weighted could improve transparency and replicability of the model’s efficacy.

**Questions:**

1. Could the authors elaborate on how they determined the weights for performance and complexity in the reward function? More detail on this could clarify the balance between the two objectives.

2。 How does TCTO perform on high-dimensional datasets with over 10,000 features? Is the pruning strategy sufficient to maintain stability without compromising feature diversity?

3. Were there any specific scenarios where TCTO’s backtracking mechanism was particularly beneficial in terms of model performance or feature diversity?

---

> ### Author Response · Authors · 2024-11-19
> **(1/2)  Discussion about the scalability and pruning strategy of TCTO**
>
> Thank you for the time and effort you have dedicated to reviewing our paper.
> We greatly appreciate your insightful comments of our work!
> Your feedback is invaluable as it confirms the **strengths of our model design, presentation and experiment**.
> The following are our detailed responses to your weaknesses and questions:
>
> **Re: weakness 1 and question 2**\
> In small-scale datasets, where the number of samples is limited and insufficient for machine learning models to learn complex patterns, feature transformation can significantly improve model performance. As a result, feature transformation is especially advantageous in small-scale datasets (see dataset description in [1][2]). That's why we didn't discuss the large-scale dataset scenarios.
>
> [1] Li L, Wang H, Zha L, et al. Learning a data-driven policy network for pre-training automated feature engineering[C]//The Eleventh International Conference on Learning Representations. 2023.\
> [2] Wang D, Xiao M, Wu M, et al. Reinforcement-enhanced autoregressive feature transformation: Gradient-steered search in continuous space for postfix expressions[J]. Advances in Neural Information Processing Systems, 2023, 36: 43563-43578.
>
> We recognize the importance of addressing the scalability of large datasets. We conducted additional experiments to evaluate its performance on large-scale datasets.\
> **TCTO can scale to large-sample datasets:** We analyzed the time consumption in Section A.3.1. Our results show that the time required for the downstream task is the primary bottleneck. In our paper, we used Random Forest (RF) implemented in scikit-learn as the downstream model. However, as the sample number increases, RF requires significantly more time. On the ALBERT dataset (425,240 samples, 78 features), each evaluation step took approximately 16 minutes.
>
> **All feature transformation methods show limited improvement on large-sample datasets:** To mitigate this issue, we switched to a more efficient downstream models using LightGBM, which offers faster training times. Table 1 demonstrates that TCTO can effectively scale by leveraging alternative models that are more efficient for large-sample datasets.
>
> Table 1: Comparison of Baseline Methods and TCTO on ALBERT
>
> |ALBERT|Original|RDG|ERG|LDA|AFAT|NFS|TTG|GRFG|DIFER|TCTO|
> |--|--|--|--|--|--|--|--|--|--|--|
> |F1-score|0.674|0.678|0.619|0.530|*|0.680|0.681|*|*|0.681|
>
> Note: * indicates that the method ran out of memory or took too long.\
> We observe that all methods exhibit limited gains, and the results indicate that feature transformation methods have limited impact on extremely large-sample datasets, which is consistent with existing work (see Table 3 in [3]). The key reason for this is that machine learning models are capable of learning latent patterns from sufficient samples, and the additional information generated by feature transformation becomes less essential.
>
> [3] Zhang T, Zhang Z A, Fan Z, et al. OpenFE: automated feature generation with expert-level performance[C]//International Conference on Machine Learning. PMLR, 2023: 41880-41901.
>
> **Pruning strategy benefit high-dimension datasets:**
> We employ two pruning strategies to preserve diversity while minimizing complexity during the initial stages and enhance exploration stability [refer to section 3.2 Roadmap Prune Strategy].
> Node-wise pruning strategy entails the identification of K nodes that show the greatest relevance to labels. For high-dimensional dataset, we employed pruning strategy on initial datasets to remove irrelevant root nodes before exploring.
>
> **TCTO can address large-dimension by employing pruning strategy:**
> We conducted experiments on high-dimensional dataset Newsgroups, which contains 13,142 samples and 61,188 features. Without pruning strategy, the evaluation step took approximately 5 minutes to complete. The results of this experiment with pruning root nodes are shown in Table 2.\
> Table 2: Comparison of Baseline Methods and TCTO on Newsgroups
>
> |Newsgroups|Original|RDG|ERG|LDA|AFAT|NFS|TTG|GRFG|DIFER|TCTO|
> |--|--|--|--|--|--|--|--|--|--|--|
> |Macro-F1|0.568|0.556|0.545|0.230|0.544|0.553|0.546|*|*|0.576|
>
> Note: * indicates that the method ran out of memory or took too long.\
> The result demonstrates that TCTO outperforms baseline methods in terms of Macro-F1. The pruning strategy helps mitigate the complexity of high-dimensional datasets and speeds up the process while maintaining performance.
> In conclusion, our experiments show that TCTO is scalable and performs well on large-sample and high-dimensional datasets when appropriate strategies (e.g., efficient downstream tasks and pruning on root nodes) are employed. We hope these clarifications address the reviewers' concerns.

---

> ### Author Response · Authors · 2024-11-19
> **(2/2) Discussion about the reward weight, backtracking mechanism and clarity of expression**
>
> **Re: weakness 3 and question 1**\
> We appreciate the reviewer’s suggestion to provide more detail on the weighting of performance and complexity in the reward function. We conducted preliminary experiments using the Airfoil dataset to determine the appropriate balance between these two rewards. We tested different reward ratios and the results are shown in Table 3.
>
> Table 3: Impact of Reward Ratio (Performance : Complexity) on Downstream Task Performance (1-RAE)
>
> |Reward ratio|0:1|0.1:0.9|0.2:0.8|0.3:0.7|0.4:0.6|0.5:0.5|0.6:0.4|0.7:0.3|0.8:0.2|0.9:0.1|1:0|
> |--|--|--|--|--|--|--|--|--|--|--|--|
> |1-RAE|0.551|0.553|0.559|0.583|0.564|0.574|0.573|0.571|0.554|0.577|0.553|
>
> As seen in the Table 3, when only the complexity reward or the performance reward is used exclusively, the performance is noticeably lower. This result suggests that while performance rewards encourage the agent to generate high-value features, overly complex features can be detrimental to the downstream task. With a balanced weight of them, the performance fluctuates slightly. Based on these preliminary results, we concluded that a ratio of 1:1 between feature quality and complexity, providing stable and reliable performance.
>
> **Re: question 3**\
> We appreciate the reviewer’s insightful question regarding the role of TCTO’s backtracking mechanism.\
> The backtracking mechanism allows the algorithm to trace back to a previously identified optimal transformation roadmap, preventing it from deviating toward suboptimal paths [refer to Section 3.2 Roadmap Prune Strategy]. This is particularly useful in scenarios when agents have explored a broad feature space. Without backtracking, the agents may become stuck in local optima or experience a significant decrease in performance. We present statistical data from Figure 10 (Airfoil Dataset), which shows the exploration step statistics for different performance intervals, both with and without the backtracking mechanism.
>
> Table 4: Comparison of Performance During Exploration with and without Backtracking
>
> |Step Count(%)|[0,0.50)|[0.5,0.51)|[0.51,0.52)|[0.52,0.53)|[0.53,0.54)|[0.54,0.55)|
> |--|--|--|--|--|--|--|
> |w.o. backtrack|30.6%|7.1%|17.2%|15.7%|13.9%|12.1%|
> |with backtrack|4.3%|2.9%|5.3%|17.5%|8.5%|12.2%|
>
> |Step Count(%)|[0.55,0.56)|[0.56,0.57)|[0.57,0.58)|[0.58,0.59)|[0.59,0.60)|[0.60,1]|
> |--|--|--|--|--|--|--|
> |w.o. backtrack|2.1%|1.1%|0.2%|0%|0%|0%|
> |with backtrack|8.8%|**20.3%**|6.8%|4.5%|4.7%|4.2%|
>
> As shown in the Table 4, when the backtracking mechanism is employed, the algorithm can frequently revert to a previously identified optimal state. In contrast, without backtracking, the agents are unable to start from a favorable state, leading to less effective exploration from the current state.\
> In summary, the backtracking mechanism significantly enhances the stability of the exploration process by allowing the algorithm to return to previous optimal states and avoid performance breakdown, ultimately leading to improved performance and more reliable feature exploration.
>
> **Re: weakness 2**\
> We reorganized and rewrite the content of cascading agents to impove the clarity, specifically:\
> (Line 196 - Line 200) Multi-agent Reinforcement Learning-based Transformation Decision:
> Reinforcement learning has proven effective in addressing complex decision-making challenges across various domains. We employ three cascading agents that collaboratively construct unary and binary mathematical transformations. These agents operate sequentially to select the optimal head cluster, mathematical operation, and operand cluster, respectively. The chosen features undergo the specified mathematical operations, resulting in the generation of new features and the creation of new nodes within the roadmap. Additional details regarding the decision-making process will be provided in Section 3.3, Cascading Reinforcement Learning Agents.
>
> (Line 313 - Line 315) Cascading Reinforcement Learning Agents: Figure 7 shows an example of the cascading agents' decision-making process. We utilize a series of cascading agents, each performing a specific task in sequential order. These agents collaborate in a step-by-step decision-making process, where the output of one agent serves as the input for the next. The first agent (head cluster agent) is responsible for selecting the head cluster, the second (operation agent) for choosing the most appropriate mathematical operation, and the third (operand cluster agent) for identifying the operand cluster. By using this cascading structure, each decision is informed by the context set by the previous agents, leading to a more efficient decision-making process. The details of each agent are as follows:
>
> Thank you once again for your time and effort. We hope that the responses provided address your concerns and sincerely hope you will reconsider the rating. If you have any further questions or require additional clarification, please do not hesitate to discuss with us.

---

> > ### Comment · Area_Chair_CLEW · 2024-11-24
> > **Please respond to rebuttal ASAP**
> >
> > Dear reviewer,
> > The process only works if we engage in discussion. Can you please respond to the rebuttal provided by the authors ASAP?

---

> ### Author Response · Authors · 2024-11-28
>
> Dear Reviewer cE2X,
>
> Thank you for the time and effort you have dedicated to reviewing our paper.
> We have revised the manuscript in accordance with your comments and have updated the PDF file accordingly. As November 27th is the final day for authors to upload a revised PDF, we have reverted the red-marked sections to black.
> In summary, the main revisions are as follows:
>
> Presentation Clarity: We have clarified the decision-making process of the cascading agents, including additional details on the workflow (Lines 195-200 and Lines 310-316). Additionally, we have rewritten the Abstract and Conclusion to enhance clarity and help readers better understand our contributions.
>
> Scalability on Large-Scale Datasets: We have analyzed the scalability of TCTO on large-scale datasets, including those with large sample sizes and high-dimensional features. This new analysis is included as an experiment in Appendix A.3.6. Based on the results, we discuss the limitations of TCTO and suggest future work to improve the approach (Lines 1138-1147).
>
> Additional Experimental Details: We have provided an explanation of how we set the reward function weights in Appendix A.3.5. Furthermore, we have clarified how we calculate the standard deviation, which is included in the table note for Table 1.
>
> Additional Baselines: We have added two recent baselines in Table 1: FETCH [1] and OpenFE [2], both of which focus on automated feature transformation.
>
> [1] Li L, Wang H, Zha L, et al. Learning a data-driven policy network for pre-training automated feature engineering[C]//The Eleventh International Conference on Learning Representations. 2023.\
> [2] Zhang T, Zhang Z A, Fan Z, et al. OpenFE: automated feature generation with expert-level performance[C]//International Conference on Machine Learning. PMLR, 2023: 41880-41901.
>
> Minor Revisions: We have made minor revisions to the presentation (Lines 33-36), Figure 1, and Formula 8 based on your comments. Additionally, we have moved the dataset source to the Appendix.
>
> We look forward to your feedback and are happy to address any further concerns.
>
> Sincerely,\
> The Authors

---

> ### Author Response · Authors · 2024-12-01
> **Kindly Reminding**
>
> Dear reviewer cE2X,
>
> We sincerely appreciate the time and effort you have invested in reviewing our manuscript. We understand that you may have been busy recently. However, as the discussion period is nearing its end (less than 48 hours remaining), we would like to kindly remind you of our rebuttal. We hope that it has addressed your concerns raised in your initial comments.
>
> We look forward to your feedback and are happy to address any further questions or concerns you may have.
>
> Best regards,\
> The Authors.

---

### Author Response · Authors · 2024-11-25
**Modification of Submission**

Dear Reviewers,\
Thank you for your valuable time and effort! Your insightful feedback has significantly helped us improve the paper. We have revised the manuscript based on your comments and updated PDF file. The revised sections are highlighted in red. The main revisions are as follows:

1. **Presentation Clarity:** We have clarified the decision-making process of the cascading agents, including additional details on the workflow (Lines 195-200 and Lines 310-316). Additionally, we have rewritten the Abstract and Conclusion to enhance clarity and help readers better understand our contributions.

2. **Scalability on Large-Scale Datasets:** We have analyzed the scalability of TCTO on large-scale datasets, including those with large sample sizes and high-dimensional features. This new analysis is included as an experiment in Appendix A.3.6. Based on the results, we discuss the limitations of TCTO and suggest future work to improve the approach (Lines 1138-1147).

3. **Additional Experimental Details:** We have provided an explanation of how we set the reward function weights in Appendix A.3.5. Furthermore, we have clarified how we calculate the standard deviation, which is included in the table note for Table 1.

4. **Additional Baselines:** We have added two recent baselines in Table 1: FETCH [1] and OpenFE [2], both of which focus on automated feature transformation.

[1] Li L, Wang H, Zha L, et al. Learning a data-driven policy network for pre-training automated feature engineering[C]//The Eleventh International Conference on Learning Representations. 2023.\
[2] Zhang T, Zhang Z A, Fan Z, et al. OpenFE: automated feature generation with expert-level performance[C]//International Conference on Machine Learning. PMLR, 2023: 41880-41901.

5. **Minor Revisions:** We have made minor revisions to the presentation (Lines 33-36), Figure 1, and Formula 8 based on your comments. Additionally, we have moved the dataset source to the Appendix.

We sincerely hope that our revisions address the concerns raised. If you have any further questions or concerns, please do not hesitate to discuss with us. We greatly look forward to receiving your feedback!

Best regards,\
The Authors

---

### Meta-Review · Area_Chair_CLEW · 2024-12-20

**Metareview:**

The paper proposes a methodology for feature transformations based on a multi agent reinforcement learning-based graph structure to maintain a roadmap of feature transformations, enabling efficient exploration and backtracking of transformation pathways. The method is evaluated on various ML datasets, showing benefits over other feature transformation methods.

Strengths:
The paper is interesting and solves a unique problem
The methodology is quite complex and it is impressive that the authors got it to perform well

Weaknesses
The paper as written is extremely hard to parse, there is *way* too much happening, there are far too many components to the system. The writing is quite hard to parse and the methodology as current constructed seems very specific and hard for others to use. For this work to meaningfully be used by others in the community, it has to be simplified and exposed in a much more clear way.

**Additional Comments On Reviewer Discussion:**

The reviewers generally felt the paper was interesting but brought up concerns about clarity, computational complexity and mention that exposition could be improved. None of the reviewers really championed the paper, and it's not clear that the author response fixed the major clarity issues in the paper.

---

### Decision · Program_Chairs · 2025-01-22

Reject